# Locally differentially private estimation of nonlinear functionals of discrete distributions

**Cristina Butucea**
CREST, ENSAE, IP Paris
Palaiseau 91120 Cedex, France
`cristina.butucea@ensae.fr`

**Yann Issartel**
CREST, ENSAE, IP Paris
Palaiseau 91120 Cedex, France
`yann.issartel@ensae.fr`

## Abstract

We study the problem of estimating non-linear functionals of discrete distributions in the context of local differential privacy. The initial data $x_1, \ldots, x_n \in [K]$ are supposed i.i.d. and distributed according to an unknown discrete distribution $p = (p_1, \ldots, p_K)$. Only $\alpha$-locally differentially private (LDP) samples $z_1, \ldots, z_n$ are publicly available, where the term 'local' means that each $z_i$ is produced using one individual attribute $x_i$. We exhibit privacy mechanisms (PM) that are sequentially interactive (i.e. they are allowed to use already published confidential data) or non-interactive. We describe the behavior of the quadratic risk for estimating the power sum functional $F_\gamma = \sum_{k=1}^K p_k^\gamma$, $\gamma > 0$ as a function of $K$, $n$ and $\alpha$. In the non-interactive case, we study two plug-in type estimators of $F_\gamma$, for all $\gamma > 0$, that are similar to the MLE analyzed by Jiao *et al.* [18] in the multinomial model. However, due to the privacy constraint the rates we attain are slower and similar to those obtained in the Gaussian model by Collier *et al.* [9]. In the sequentially interactive case, we introduce for all $\gamma > 1$ a two-step procedure which attains the parametric rate $(n\alpha^2)^{-1/2}$ when $\gamma \geq 2$. We give lower bounds results over all $\alpha$-LDP mechanisms and all estimators using the private samples.

## 1 Introduction

Information theoretic measures have become of utmost importance and extensively used in information theory, image processing, physics, genetics and more recently in machine learning and statistics. Such functionals of probability distributions are useful to design estimators, to choose most informative features in different algorithms, to test identity or closeness of distributions. Popular such measures are the power-sum, the entropy and more general Rényi entropies of discrete distributions.

In this paper, we are interested in estimating the power sum functional $F_\gamma(p)$ of a discrete distribution $p = (p_1, ..., p_K)$:

$$F_\gamma(p) = \sum_{k=1}^K p_k^\gamma, \quad \text{with power } \gamma \in (0, \infty).$$

This instance of information measure has a tight connection with the famous Rényi entropy $H_\gamma$ via the formula $H_\gamma = \frac{\log F_\gamma}{1-\gamma}$.

In the statistical literature, smooth nonlinear functionals are often reduced via Taylor expansion to several functionals of the type $F_\gamma$ for positive integer values of $\gamma$ (see monographs like e.g. [14] ). For example, the entropy $H(p) = \sum_{j=1}^K p_j \log(1/p_j)$ of the probability distribution $p$ with finite support and probabilities bounded away from 0 can be approximated via a Taylor expansion to a linear combination of $F_\gamma$ for integer values of $\gamma$ and estimated at parametric rate of $1/\sqrt{n}$.

35th Conference on Neural Information Processing Systems (NeurIPS 2021).

Symmetric functions of $p_1, \ldots, p_K$, i.e. functions of at most $K$ variables that are permutation invariant, play an important role in deep learning, e.g. [30], [23]. Such functions can be written as polynomials of functionals $F_\gamma$ for integer values of $\gamma$ and estimated using our procedures.

Another important application of such functionals is testing identity or closeness of distributions. For example, let us consider uniformity testing that is the null hypothesis is $H_0 : p_j = 1/K$ for all $j$ from 1 to $K$ against the alternative hypothesis $p$ is not the uniform distribution. Suppose the distance measuring how far $p$ is from the uniform distribution is the Hellinger distance $\sum_{k=1}^{K}(\sqrt{p_k} - 1/\sqrt{K})^2$. Therefore a test procedure will proceed by estimating this Hellinger distance using the sample and this involves estimating $F_{1/2}$ for $\gamma = 1/2$. Thus, an uniformity test procedure will be based on the estimator of this discrepancy, and similarly for identity or closeness tests. More generally, when the distance between two probability distributions is evaluated by a discrepancy or a distance, functionals $F_\gamma$ naturally appear in their expression. Note however that testing rates may differ from the estimation rates of the discrepancy as is the case for the $L_1$ distance where [26] showed that several tests procedures must be aggregated in order to attain better rates for testing. We stress the fact that testing is a different problem from learning the functional.

## 1.1 Plug-in Approach

In the standard statistical setup (also called multinomial setting), the goal is to estimate the power sum functional $F_\gamma$ based on $n$ i.i.d. samples $x_1, ..., x_n$ following an unknown discrete distribution $p = (p_1, \ldots, p_K)$ with alphabet size $K$. A commonly-used approach to this problem is the plug-in approach, which amounts to using an estimate $\hat{p}$ of the parameter $p$ in order to build an estimator $F_\gamma(\hat{p})$ of the functional $F_\gamma(p)$. The resulting plug-in estimator actually corresponds to the so-called maximum likelihood estimator (MLE) when the estimate $\hat{p} = (\hat{p}_k)_{k \in [K]}$ is defined as the empirical distribution $\hat{p}_k = \frac{1}{n} \sum_{i=1}^{n} \mathbb{1}_{x_i = k}$ (i.e. the average counts in the $k^{\text{th}}$ box after binning). This approach is not only intuitive and simple, but is also theoretically well grounded, as it is asymptotically efficient for finitely supported probabilities (finite $K$), and is non-asymptotically nearly optimal for possibly increasing $K$ [18] (for details, see the related literature below). A natural question that we investigate in this paper is whether such plug-in type approach still performs well in a non-standard statistical setup where a constraint of privacy is imposed on the observed data.

In the standard setup, estimation of power-sum functionals was studied for the widely spread MLE estimator in [18] where the authors found that its maximal quadratic risk (also called worst case squared error risk) is sub-optimal only by some logarithmic factor. The estimation rates have then been tightened to minimax optimal bounds using the best polynomial approximation of the power function [17], [29]. Following the chronology of this literature, our study in a non-standard privacy setup starts with an analogue of this MLE, with a twofold purpose: (i) to highlight an important difference between the LDP setting and the standard (non private) setting: practical methods performing well in the non private case should not be systematically transferred to the private settings, a good estimator in the first setup not being necessarily a good estimator in the latter setup; (ii) to show the regimes where plug-in type estimators fail and the estimation problem is delicate, thus setting benchmarks for future work on functional estimation in the LDP setting, as well as more involved private settings.

## 1.2 Differential Privacy

Keeping sensitive data $x_1, \ldots, x_n$ private is a major concern in the modern area of Big Data. For example, $x_1, \ldots, x_n$ may be personal health or financial data of $n$ participants to a survey. *Differential privacy* (DP) [12] has prevailed in the recent literature as a convenient approach to randomize samples with a control of the amount $\alpha > 0$ of privacy introduced. The randomized samples $z_1, \ldots, z_n$, also known as private samples, are provided to the statisticians who want to extract information on the underlying distribution $p$ of the initial data $x_1, \ldots, x_n$. *Global or central DP* allows simultaneous treatment of the whole initial sample in order to produce the privatized random variables. In contrast, *Local differential privacy* (LDP) is a stronger setup of privacy where no one has access to all sensitive data $x_1, \ldots, x_n$ (not even a trusted curator or third party to handle the privatization), but each individual $i$ has access to one $x_i$.

It is quite popular now that privacy, in particular local differential privacy, comes at the cost of slower rates of learning in many estimation problems. For example, estimation of the probability density is known to be achieved with slower rates in the minimax sense under $\alpha$-LDP constraints, see [10],

[28], [24] and [4]. There is a rush nowadays to better understand when the loss is unavoidable and to show that such loss is optimal over all privacy mechanisms and procedures at hand.

In this LDP setup, there are major distinctions according to the way the privacy mechanisms use the available information, that is non-interactive, sequentially interactive or fully interactive setups. When using *non-interactive* mechanisms, each private sample $z_i$ is produced by a privacy mechanism $Q_i$ that has access only to one sensitive sample $x_i$. They are arguably the simplest mechanisms. A richer class of mechanisms are the so-called *sequentially interactive* mechanisms where each agent $i$ is allowed to incorporate into its private mechanism $Q_i$ the already privatized data $z_1, \ldots, z_{i-1}$ of other agents, along with $x_i$. It is known e.g. [5] that sequentially interactive methods attain much faster rates than the non-interactive mechanisms in some inference problems. This has also been proved for identity testing of discrete distributions in [3]. Finally, the *fully interactive* privacy mechanisms are allowed to use one $x_i$ several times, together with all publicly available randomized $z$'s. In this case, the natural question is how many randomizations are necessary in order to acquire the desired amount of information. It has been proved that there exist separations between fully and sequentially interactive procedures, e.g. [20] and [21], as well as separations between fully interactive mechanisms and global DP, e.g. [8]. We do not consider fully interactive mechanisms in the current paper (neither in the upper, nor in the lower bounds).

Understanding the relative power of interactivity is a crucial question in the LDP setting. In this line of work, we study the estimation of the power sum functional $F_\gamma$ under the constraints that the available data $z_1, \ldots, z_n$ stem from non-interactive and sequentially interactive privacy mechanisms, respectively.

## 1.3 Contributions

In the $\alpha$-LDP setting we propose three estimators of the power sum function $F_\gamma(p)$ based on the $n$ randomized observations. The performance of any estimator $\hat{F}_\gamma$ is controlled by proving upper bounds on its non-asymptotic quadratic risk $\mathbb{E}[(\hat{F}_\gamma - F_\gamma)^2]$, with an explicit dependence on the parameters of the problem: the power $\gamma$, the alphabet size $K$, the sample size $n$ and the amount of privacy $\alpha$.

Our first contribution is a tight characterization of the quadratic risk of the *plug-in estimator*, an analogue of the plug-in MLE (discussed above). Unfortunately, its risk grows rapidly with the alphabet size $K$, thus showing that the plug-in approach is by far not optimal for large $K$. This contrasts with the good performance of the plug-in in the standard statistical setup [18]. Thus, good estimators in the standard setup are not necessarily to be used as such in the LDP setting.

Our second contribution is a correction of this plug-in estimator by truncating the small probabilities $p_k$. The induced procedure, called *thresholded estimator* in the sequel, performs significantly better than the plug-in estimator, typically when the alphabet size $K$ is large. We emphasize that this improvement is important, since the risk of this thresholded estimator is (almost) independent of the support size $K$ for large $K$, unlike the risk of the plug-in estimator. It is therefore different from the literature on functional estimation (in the standard setting) where the improvements in the risk of plug-in estimators are often of magnitude of logarithmic factors.

The privatized data $z_1, \ldots, z_n$ used by both our plug-in and thresholded estimators are generated by a simple Laplace non-interactive mechanism. In contrast, our third contribution is a two-step procedure based on a sequentially interactive mechanism. The definition of this two-step procedure heavily relies on the plug-in estimator, and thus can be seen as a refinement of the plug-in approach. Such a sequentially interactive method was studied in [5] for the particular power sum functional $F_2$ ($\gamma = 2$), also called quadratic functional, in a continuous setup (where the probability distribution is a smooth function on $[0, 1]$). By allowing to encode information from previous observations $z_1, \ldots, z_{i-1}$ into new released data $z_i$, this sequentially interactive procedure achieves faster rates than the thresholded estimator, when $K$ is large and $\gamma > 1$, though this improvement is only of a logarithmic factor. Unfortunately, this sequentially interactive procedure is only defined for $\gamma > 1$, and has slower rates than the thresholded and plug-in estimators when $K$ is small. Accordingly, none of our three estimators is overall better than the others. Table 1 presents upper bounds on the maximal quadratic risks of these three estimators. The choice of estimators therefore depends on regimes (i.e. values of $K, \gamma$), and our fastest rates attained by some combination of these three estimators are written in Corollary 2.5.

Table 1: Upper bounds on maximal quadratic risks of three procedures

| | $\gamma \in (0,1)$ | $\gamma > 1$ |
|---|---|---|
| Plug-in estimator (*Non-Interactive PM*) | $\frac{K^2}{(\alpha^2 n)^\gamma}$ | $\frac{K^2}{(\alpha^2 n)^\gamma} + \frac{K^{3-2\gamma} \vee 1}{\alpha^2 n} \quad := R_1$ |
| Thresholded estimator (*Non-Interactive PM*) | $K^{2(1-\gamma)} \wedge \frac{K^2}{(\alpha^2 n)^\gamma}$ | $\min\{R_1, R_2\}. \log(Kn)^\gamma$ |
| Two-step procedure (*Interactive PM*) | | $\frac{1}{(\alpha^2 n)^{\gamma-1}} + \frac{1}{\alpha^2 n} \quad := R_2$ |

We finally give lower bounds on the maximal quadratic risk, over all estimators and all (non-interactive and sequentially interactive) privacy mechanisms. These lower bounds are optimal for $\gamma \geq 2$ as they match our fastest rates. Unfortunately for $\gamma \in (0,2)$, a gap of a factor $K^\gamma$ remains between our lower bounds and fastest rates. All proofs are in [6].

## 1.4 Related Literature

To the best of our knowledge, estimating non linear functionals under global DP has been considered by [2]. The authors estimate the entropy, the support size and the support coverage of a discrete distribution in the context of global differential privacy. In each setup they show that the cost of privacy is relatively small when compared to the standard (non private) setup. Their upper bounds allow to quantify the amount of privacy that we can have without deteriorating the estimation rates.

We study for the first time the estimation of the nonlinear power-sum functional $F_\gamma$ for any real $\gamma > 0$ in the context of LDP. The case $\gamma = 2$ has been considered in the LDP setup, for smooth distributions in [5], and for testing discrete distributions in [3]. In [5], the association of a plug-in estimator and Laplace mechanism is optimal among all non-interactive mechanisms, whereas a sequentially interactive procedure improves dramatically the minimax rates. We will also try to further understand when such phenomena hold for different values of $\gamma$, in particular when the functional $F_\gamma$ is less smooth for $\gamma < 2$.

In non-private settings, it has been shown different behaviors for estimating $F_\gamma$ according to the observations scheme, namely faster rates are attained in the multinomial setup [18] than in the Gaussian vector model [9] − see details below. We will see that due to the LDP setup our rates are similar to those proved in the Gaussian vector model by [9], though our plug-in type estimator is an analogue of the MLE analyzed in the multinomial setting by [18].

*Standard (or multinomial) setting:* When the r.v. $x_1, ..., x_n$ are observed, [18] show that the maximal quadratic risk of the maximum likelihood estimator (MLE) in estimating $F_\gamma$, $\gamma > 0$, is

$$\frac{K^2}{n^{2\gamma}}\mathbf{1}_{\gamma \in (0,1)} + \frac{K^{2(1-\gamma)}}{n}\mathbf{1}_{\frac{1}{2} < \gamma < 1} + n^{-2(\gamma-1)}\mathbf{1}_{1 < \gamma < \frac{3}{2}} + \frac{1}{n}\mathbf{1}_{\gamma \geq \frac{3}{2}}.$$

In this model, the MLE of $p_k$ is the average of the $\mathbf{1}_{x_i=k}$ over $i$ from 1 to $n$. These results entail that the MLE achieves the minimax rate $n^{-1}$ when $\gamma \geq 3/2$. However, when the regularity $\gamma$ decreases below $3/2$, the rates of the MLE get slower. In particular, the MLE does not achieve the minimax rates of estimation of $F_\gamma$ when $0 < \gamma < 3/2$. For $0 < \gamma < 1$, the difficulty of estimation increases, especially if the alphabet size $K$ is large, which may even prevent from a consistent estimation of $F_\gamma$ (i.e. from the risk of the MLE to converge to zero). More precisely, the MLE consistently estimates $F_\gamma$, $0 < \gamma < 1$, if and only if, the sample size is large enough to satisfy $n \gg K^{1/\gamma}$.

For the values of $\gamma < 3/2$, the minimax rates in estimating $F_\gamma$ roughly correspond to the performance of the MLE (written above) with $n$ replaced by $n \log n$ [17]. These faster rates have been attained through a different estimator that uses best polynomial approximation of the functional $p_k^\gamma$ for small values of $p_k$, which allows to gain a log factor in the bias. This method has been largely employed in the literature since the pioneer work by [22], in the Gaussian white noise model. The best polynomial approximation technique has been used for estimating the $\mathbb{L}_1$ norm in a Gaussian setting [7], the $\mathbb{L}_r$ norm in the Gaussian white noise model [15], the entropy of discrete distributions [29], the entropy of Lipschitz continuous densities [16]. In the multinomial setting, [1] studied the estimation of the

Rényi entropy of order $a > 0$ and [13] extended these results to more general additive functionals of the form $\sum_{k=1}^{K} \varphi(p_k)$, for 4-times differentiable functions $\varphi$.

*Gaussian vector setting:* the observations are $y_i = \theta_i + \epsilon \cdot \xi_i$, $i = 1, \ldots, K$ where $(\theta_1, \ldots, \theta_K)$ are unknown parameters, $\epsilon > 0$ is known, and $\xi_i$ are i.i.d. standard Gaussian random variables. The authors of [9] show that there exists an estimator based on best polynomial approximation with the following error bound − see (14) in the proof of their Theorem 1 −

$$\mathbb{E}\left[\left(\hat{F}_\gamma - F_\gamma\right)^2\right] \lesssim \frac{\epsilon^{2\gamma} K^2}{\log^\gamma(K)} + \frac{\epsilon^2}{\log(K)} \left(\sum_{i=1}^{K} \theta_i^{\gamma-1}\right)^2 \mathbb{1}_{\gamma>1} + \epsilon^2 \sum_{i=1}^{K} \theta_i^{2\gamma-2} \mathbb{1}_{\gamma>1} \qquad (1)$$

for any real $\gamma > 0$. When $\gamma \in (0, 1)$, this bound is equal to $\epsilon^{2\gamma} K^2 / \log^\gamma(K)$ and turns out to be minimax optimal [9]. Besides, the authors show that better rates can be achieved if $\gamma$ is an integer. Namely, there exists an estimator with the following error bound, see Theorem 2 in [9],

$$\mathbb{E}\left[\left(\hat{F}_\gamma - F_\gamma\right)^2\right] \lesssim \epsilon^{2\gamma} K + \epsilon^2 \|\theta\|_{2\gamma-2}^{2\gamma-2} \qquad (2)$$

for any integer $\gamma \geq 1$. In this special case of an integer $\gamma$, the rate (2) is achieved by a simple estimator that has no bias.

## 2 Estimators and results

We aim at estimating the power sum functional $F_\gamma(p) = \sum_{k=1}^{K} p_k^\gamma$, for $\gamma \in (0, \infty)$, in the LDP setting where we only have access to privatized versions $z_1, \ldots, z_n$ of the sensitive original data $x_1, \ldots, x_n \overset{\text{i.i.d.}}{\sim} p$. The sensitive random variables $x_1, ..., x_n$ are i.i.d. distributed according to the discrete density model $\mathbb{P}(x_i = k) = p_k$ for $k = 1, \ldots, K$ where the unknown parameter $p = (p_1, \ldots, p_K)$ belongs to the set $\mathcal{P}_K = \{p \in [0,1]^K : \sum_{k=1}^{K} p_k = 1\}$ of discrete distributions with alphabet size $K$. The $x_1, \ldots, x_n$ are not publicly available; instead they are used as inputs into a privacy mechanism (PM), in order to produce available sanitized observations $z_1, ..., z_n$. We use the sanitized observations to estimate the functional $F_\gamma$.

Notation: The symbol $x \lesssim_\gamma y$ means that the inequality $x \leq C_\gamma y$ holds for some constant $C_\gamma$ depending only on $\gamma$. We denote $\min\{x, y\}$ by $x \wedge y$, and $\max\{x, y\}$ by $x \vee y$.

### 2.1 LDP setup

The privacy mechanism PM also known as channel or randomization, is submitted to the constraint that it is $\alpha$-locally differentially private (LDP) for some $\alpha > 0$. This means that the PM generates the private samples $z_i$ using a conditional distribution $Q_i(\cdot | x_i, z_1, ..., z_{i-1})$ such that

$$\sup_{z_1, ..., z_{i-1}, x_i, x_i'} \frac{Q_i(\cdot | x_i, z_1, ..., z_{i-1})}{Q_i(\cdot | x_i', z_1, ..., z_{i-1})} \leq e^\alpha, \qquad \text{for all } i = 1, ..., n, \qquad (3)$$

with the convention that $\{z_1, \ldots, z_{i-1}\}$ is the empty set for $i = 1$. These PM are called sequentially interactive, as each $Q_i$ is allowed to use previously published samples $z_1, ..., z_{i-1}$. We are also interested in the sub-class of non-interactive PM which are not allowed to use previous released data. A non-interactive PM generates each $z_i$ only accessing $x_i$, via a conditional distribution of the form $Q_i(\cdot | x_i)$. We assume from now on that the level $\alpha$ of privacy belongs to $(0, 1)$ and satisfies $\alpha^2 n \geq 1$.

### 2.2 Non-interactive privacy mechanism (NI PM)

We introduce a non-interactive PM, denoted by $Q^{(NI)}$. Given the original data $x_i$, individual $i$ generates a random vector $z_i = (z_{i1}, \ldots, z_{iK})$ using the Laplace non-interactive privacy mechanism $Q^{(NI)}$ defined by

$$Q^{(NI)}: \qquad z_{ik} = \mathbf{1}_{\{x_i=k\}} + \frac{\sigma}{\alpha} \cdot w_{ik}, \qquad k = 1, \ldots, K, \qquad (4)$$

where the $w_{ik}$ are i.i.d. Laplace distributed with density $f^w(x) = \frac{1}{2} \exp(-|x|)$. Note that $w_{ik}$ are all centered, with variance 2. Setting $\sigma = 2$, one can readily check that the channel $Q^{(NI)}$ above is an $\alpha$-LDP non-interactive mechanism, see [3] and the references therein. We denote the mean of the privatized observations in the $k^{\text{th}}$-box by $\hat{z}_k = \frac{1}{n} \sum_{i=1}^{n} z_{ik}$.

## 2.3 Plug-in estimator based on NI PM

The first estimator we introduce is an analogue of the MLE [18] discussed in the introduction. It uses an unbiased estimation of the parameter $p$, by averaging the privatized samples in each bin, thus using the $\hat{z}_k$, $k = 1, \ldots, K$. The resulting estimator $\hat{F}_\gamma$ of $F_\gamma$ is actually not a MLE in our privatized context, hence we call it plug-in estimator. The plug-in estimator uses the privatized data produced by the NI PM above, and then estimates separately each term $F_\gamma(k) = p_k^\gamma$ of the functional $F_\gamma = \sum_{k=1}^K F_\gamma(k)$ as follows:

$$\hat{F}_\gamma = \sum_{k=1}^K \hat{F}_\gamma(k) \ , \ \text{with} \ \hat{F}_\gamma(k) = \left(T_{[0,2]}\left[\hat{z}_k\right]\right)^\gamma \ , \tag{5}$$

where $T_{[0,2]}\left[\cdot\right]$ is the clipping operation defined by $T_{[0,2]}\left[y\right] = (y \vee 0) \wedge 2$. As $\hat{z}_k$ is an unbiased estimator of $p_k$ with fluctuations of order $(\sqrt{p_k(1-p_k)}/n) + (\sigma/\sqrt{\alpha^2 n})$, the quantity $\tau = c/\sqrt{\alpha^2 n}$, where $c \geq 1$ is a numerical constant, can be seen as a threshold that is just above the noise level in the available data. Write $p^{\geq \tau}$ (respectively $p^{<\tau}$) the vector containing the thresholded values of $p$: $p_k \cdot \mathbf{1}(p_k \geq \tau)$ (respectively $p_k \cdot \mathbf{1}(p_k < \tau)$) for $k \in [K]$.

**Theorem 2.1.** *For any $\gamma > 0$ and $p \in \mathcal{P}_K$, the quadratic risk of the estimator (5) is bounded by*

$$\mathbb{E}_p\left[(\hat{F}_\gamma - F_\gamma)^2\right] \lesssim_\gamma \frac{K^2}{(\alpha^2 n)^\gamma} + \mathbf{1}_{\{\gamma \geq 2\}} \frac{\|p^{\geq \tau}\|_{\gamma-2}^{2(\gamma-2)}}{(\alpha^2 n)^2} + \mathbf{1}_{\{\gamma \geq 1\}} \frac{\|p^{\geq \tau}\|_{2\gamma-2}^{2\gamma-2}}{\alpha^2 n} \ , \tag{6}$$

*and thus is uniformly bounded over all $p \in \mathcal{P}_K$ by*

$$\sup_{p \in \mathcal{P}_K} \mathbb{E}\left[(\hat{F}_\gamma - F_\gamma)^2\right] \lesssim_\gamma \frac{K^2}{(\alpha^2 n)^\gamma} + \mathbf{1}_{\{\gamma \geq 1\}} \frac{(\tau^{-1} \wedge K)^{3-2\gamma} \vee 1}{\alpha^2 n} \ .$$

This upper bound on the quadratic risk of the plug-in estimator grows quadratically with the alphabet size $K$, even for arbitrarily large $K$. This poor performance is unfortunately inherent to the plug-in estimator, as shown by the lower bound in Proposition 2.2. Accordingly, we will correct the plug-in estimator in the next section to reduce the effect of the alphabet size $K$ on the risk.

**Proposition 2.2.** *For any $\gamma > 0$, $\gamma \neq 1$, and integer $K \geq 2$, the maximal quadratic risk of $\hat{F}_\gamma$ is asymptotically bounded from below by*

$$\sup_{p \in \mathcal{P}_K} \mathbb{E}\left[(\hat{F}_\gamma - F_\gamma)^2\right] \gtrsim_\gamma \frac{K^2}{(\alpha^2 n)^\gamma} \qquad \text{as } n \to \infty \ .$$

The maximal quadratic risk of the plug-in estimator has therefore a quadratic dependence on the alphabet size $K$, and thus it is necessary and sufficient to have $n \gg K^{2/\gamma} \alpha^{-2}$ initial observations for this estimator to be consistent (in the sense that its risk converges to zero). This rate is slower than that of the MLE in the standard setup without privacy constraint [18], though both our plug-in estimator and the MLE have similar definitions based on the plug-in principle. Indeed, the MLE of [18] is less sensitive to the support size $K$ than our plug-in estimator, as the MLE is consistent regardless of the alphabet size $K$, as soon as $\gamma > 1$. In addition, when $\gamma \in (0, 1)$, the MLE is consistent if, and only if, $n \gg K^{1/\gamma}$. This better performance of the MLE can be explained by the fact that [18] benefits from direct observations $\hat{x}_k := \frac{1}{n}\sum_{i=1}^n \mathbf{1}_{\{x_i=k\}}$ that have non-homogeneous fluctuations, i.e. a variance $\mathrm{Var}(\hat{x}_k) = p_k(1-p_k)/n$ that scales with the signal $p_k$. By contrast, our plug-in estimator employs privatized observations $\hat{z}_k$ resulting from the Laplace PM in (4), which have nearly homogeneous fluctuations with a variance $\mathrm{Var}(\hat{z}_k)$ that scales with $\tau^2 \asymp (\alpha^2 n)^{-1}$ regardless of $k \in [K]$. Thus, both situations are different for small $p_k$, which typically occurs when $K$ is large since $\sum_{k=1}^K p_k = 1$.

*Link with the Gaussian vector model.* The observations released by the Laplace PM in (4) are in fact close to the observations of the Gaussian vector model [9] and, as a consequence, our plug-in estimator achieves similar rates to those obtained in [9]. Indeed, our observations $(\hat{z}_k)_{k \in [K]}$ can be seen as homoskedastic random variables having common characteristics to the Gaussian random variables in [9] with variance $\epsilon_0^2 = 1/(\alpha^2 n)$. Let us check that our upper bound (6) implies, up to log factors, the same bound as (1) from [9]. (The logarithmic gap comes from the fact that we do not use

the best polynomial approximation, unlike [9] − see the introduction for details.) The square root of the second term $\|p^{\geq \tau}\|_{\gamma-2}^{\gamma-2}/(\alpha^2 n)$ in (6) is bounded from above by

$$\frac{\sum_{k=1}^{K} p_k^{\gamma-2}\mathbb{1}_{\{p_k>\tau\}}}{\alpha^2 n} \leq \sum_{k=1}^{K} \frac{p_k^{\gamma-1}}{\sqrt{\alpha^2 n}}\frac{p_k^{-1}\mathbb{1}_{\{p_k>\tau\}}}{\sqrt{\alpha^2 n}} \leq \sum_{k=1}^{K}\frac{p_k^{\gamma-1}c^{-1}}{\sqrt{\alpha^2 n}} = \frac{\|p\|_{\gamma-1}^{\gamma-1}c^{-1}}{\sqrt{\alpha^2 n}}$$

which corresponds to the square root of the second term of (1), up to a log factor. Thus, our bounds are similar to those in [9] for a specific variance $\epsilon_0^2$. Note the difference that we impose the constraint $p \in \mathcal{P}_K$, whereas [9] consider all $p \in \mathbb{R}^K$.

## 2.4 Thresholded plug-in estimator based on NI PM

Our objective is to reduce the effect of the alphabet size $K$ on the risk of the plug-in estimator. A natural solution is to not estimate the small probabilities $p_k$, as their contribution to the functional $F_\gamma$ is relatively weak, while they may add much fluctuations to the plug-in estimator $\hat{F}_\gamma = \sum_{k=1}^{K} \hat{F}_\gamma$. Accordingly, we introduce a refinement of the plug-in estimator, called *thresholded plug-in estimator* $\overline{F}_\gamma$, which does not estimate all components $p_k^\gamma$ of the sum $F_\gamma = \sum_k p_k^\gamma$. Similar to the plug-in estimator, this estimator $\overline{F}_\gamma$ uses the private sample $(z_{ik})_{i\in[n],k\in[K]}$ produced by the non-interactive PM in (4).

The definition of $\overline{F}_\gamma$ is split in two cases, $\gamma \in (0, 1)$ and $\gamma > 1$. When $\gamma \in (0,1)$, we simply set $\overline{F}_\gamma := \mathbb{1}_{K \leq \tau^{-1}}\hat{F}_\gamma$, meaning that $\overline{F}_\gamma$ is equal the trivial estimator 0 if $K \geq \tau^{-1}$, and to the plug-in estimator $\hat{F}_\gamma$ otherwise.

For $\gamma > 1$, the thresholded estimator $\overline{F}_\gamma$ uses the plug-in estimator only on the significant components $p_k^\gamma$ of the sum $F_\gamma = \sum_k p_k^\gamma$, via a truncation of the small probabilities $p_k$. This two-step procedure first detects the significant probabilities $p_k$ that are above the threshold $\tau = c/\sqrt{\alpha^2 n}$, and then estimates the part of the functional $F_\gamma$ induced by those $p_k$. Assume that the sample size is $2n$ for convenience, and split the data in two samples $x^{(1)} = (x_1^{(1)}, \ldots, x_n^{(1)})$ and $x^{(2)} = (x_1^{(2)}, \ldots, x_n^{(2)})$. The individuals owning the data $z^{(s)}$, $s = 1, 2$, use the non-interactive mechanism (4) which generates $z_i^{(s)} = (z_{i1}^{(s)}, \ldots, z_{iK}^{(s)})$ for $i = 1, \ldots, n$. Denote the two sample means of the $k^{\text{th}}$ bin by $\hat{z}_k^{(s)} = \frac{1}{n}\sum_{i=1}^{n} z_{ik}^{(s)}$, $s = 1, 2$. We use the $(\hat{z}_k^{(1)})_k$ to detect large values of the underlying probabilities as follows. For each $k \in [K]$, if $\hat{z}_k^{(1)}$ is strictly smaller than the the empirical threshold $\hat{\tau}$,

$$\hat{\tau} := 192\sigma\sqrt{\frac{\log(Kn)}{\alpha^2 n}} \ ,$$

then we do not estimate $F_\gamma(k)$. The threshold $\hat{\tau}$ is chosen larger than $\tau$ by a logarithmic factor, only to have high probability concentration for the $(\hat{z}_k^{(1)})_k$ around their means $(p_k)_k$. Otherwise, when $\hat{z}_k^{(1)} \geq \hat{\tau}$, we estimate $F_\gamma(k)$ using the same plug-in estimator as in (5). This gives the following estimator $\overline{F}_\gamma$ of $F_\gamma$, $\gamma > 1$,

$$\overline{F}_\gamma = \sum_{k=1}^{K}\overline{F}_\gamma(k) \ , \ \text{ with } \overline{F}_\gamma(k) := \left(T_{[0,2]}\left[\hat{z}_k^{(2)} \cdot \mathbf{1}_{\hat{z}_k^{(1)}\geq\hat{\tau}}\right]\right)^\gamma \ , \tag{7}$$

where the $(\hat{z}_k^{(2)})_k$ are used to estimate the functional. The independence between $(\hat{z}_k^{(1)})_k$ and $(\hat{z}_k^{(2)})_k$ allows us to avoid cumbersome statistical dependencies between the two stages of the procedure.

**Theorem 2.3.** *For any integers $K, n$ satisfying $n \geq 2\log(K)$, the quadratic risk of $\overline{F}_\gamma$ is uniformly bounded over all $p \in \mathcal{P}_K$ by*
1) $\gamma \in (0, 1)$:

$$\sup_{p\in\mathcal{P}_K} \mathbb{E}\left[(\overline{F}_\gamma - F_\gamma)^2\right] \lesssim_\gamma K^{2(1-\gamma)} \wedge \frac{K^2}{(\alpha^2 n)^\gamma} \ .$$

2) $\gamma > 1$:

$$\sup_{p\in\mathcal{P}_K} \mathbb{E}\left[(\overline{F}_\gamma - F_\gamma)^2\right] \lesssim_\gamma \left(\frac{(\log(Kn))^{\gamma-1}}{(\alpha^2 n)^{\gamma-1}} + \frac{1}{\alpha^2 n}\right) \wedge \left(\frac{K^2(\log(Kn))^\gamma}{(\alpha^2 n)^\gamma} + \frac{K^{3-2\gamma}\vee 1}{\alpha^2 n}\right).$$

Compared to the plug-in estimator performance, the quadratic risk of the thresholded estimator $\overline{F}_\gamma$ is much less sensitive to the alphabet size $K$ when $\gamma > 1$ and $K$ is large. Indeed, our bound on this risk is a minimum between two error terms where the left-term only depends on $K$ logarithmically. However, the new logarithmic factors that come from our use of high probability concentration are not satisfactory. We remove them in the next section, replacing the current non-interactive PM with a sequentially interactive PM.

## 2.5 Two-step procedure based on sequentially interactive PM

A sequentially interactive PM is allowed to use the prior released (sanitized) data to encode our present knowledge in the new released data. The idea is to rewrite the functional $F_\gamma$ as $\sum_{k=1}^{K} p_k \cdot F_{\gamma-1}(k)$, so that we first compute the plug-in estimator $\hat{F}_{\gamma-1}(k)$ of $F_{\gamma-1}(k)$, and then build on $\hat{F}_{\gamma-1}$ to estimate the functional $F_\gamma$. In this two step procedure, half of the sample is released via the Laplace mechanism (4) and is used for computing $\hat{F}_{\gamma-1}(k)$, then the other half is released through a sequentially interactive PM encoding the information from $\hat{F}_{\gamma-1}(k)$. This procedure is only applicable for $\gamma > 1$ since it requires to compute $\hat{F}_{\gamma-1}$. The sequentially interactive PM we consider here is similar to the ones studied for the specific functional $F_2$ ($\gamma = 2$) in continuous setting [5] and identity testing [3].

Assuming that the sample size is $2n$ for convenience, we split the data in two groups $x^{(1)} = (x_1^{(1)}, \ldots, x_n^{(1)})$ and $x^{(2)} = (x_1^{(2)}, \ldots, x_n^{(2)})$. The individuals owning the data $x^{(1)}$ use the non-interactive mechanism (4), i.e.

$$Q^{(NI)}: \qquad z_{ik}^{(1)} = \mathbf{1}_{\{x_i^{(1)} = k\}} + \frac{\sigma}{\alpha} \cdot w_{ik}, \qquad k = 1, \ldots, K,$$

which generates $z_i^{(1)} = (z_{i1}^{(1)}, \ldots, z_{iK}^{(1)})$ for $i = 1, \ldots, n$. Denote this first sample by $z^{(1)} = (z_1^{(1)}, \ldots, z_n^{(1)})$. These sanitized data allow us to estimate $F_{\gamma-1}(k) = (p_k)^{\gamma-1}$ using the plug-in estimator (5), i.e.

$$\hat{F}_{\gamma-1}^{(1)}(k) = \left( T_{[0,2]} \left[ \frac{1}{n} \sum_{i=1}^{n} z_{ik}^{(1)} \right] \right)^{\gamma-1} =: \left( T_{[0,2]} \left[ \hat{z}_k^{(1)} \right] \right)^{\gamma-1} .$$

We then design an estimator of $\sum_{k=1}^{K} p_k \hat{F}_{\gamma-1}^{(1)}(k)$ which can be seen as a proxy of $F_\gamma$. This is possible with the following sequentially interactive mechanism that encodes prior information $\hat{F}_{\gamma-1}^{(1)}$ in the released data $z_i^{(2)}$:

$$Q^{(I)}: \qquad z_i^{(2)} = \pm z_\alpha, \text{with probability } \frac{1}{2} \left( 1 \pm \frac{1}{z_\alpha} \hat{F}_{\gamma-1}^{(1)}(x_i^{(2)}) \right)$$

where $z_\alpha = 2^{\gamma-1} \frac{e^\alpha + 1}{e^\alpha - 1}$, and with the following sequentially interactive estimator

$$\widetilde{F}_\gamma = \frac{1}{n} \sum_{i=1}^{n} z_i^{(2)} .$$

Denoting the second sample by $z^{(2)} = (z_1^{(2)}, \ldots, z_n^{(2)})$, it is easy to see that the privatized sample $(z^{(1)}, z^{(2)})$ satisfies (3) and thus is $\alpha-$LDP .

**Theorem 2.4.** *For any $\gamma > 1$ and $p \in \mathcal{P}_K$, the quadratic risk of the sequentially interactive estimator $\widetilde{F}_\gamma$ is bounded by*

$$\mathbb{E}\left[ (\widetilde{F}_\gamma - F_\gamma)^2 \right] \lesssim_\gamma \frac{1}{(\alpha^2 n)^{(\gamma-1) \wedge 1}} + \mathbf{1}_{\{\gamma \geq 3\}} \frac{\|p^{\geq \tau}\|_{\gamma-2}^{2(\gamma-2)}}{(\alpha^2 n)^2} + \mathbf{1}_{\{\gamma \geq 2\}} \frac{\|p^{\geq \tau}\|_{2\gamma-2}^{2\gamma-2}}{\alpha^2 n},$$

*and thus is uniformly bounded over all $p \in \mathcal{P}_K$ by*

$$\sup_{p \in \mathcal{P}_K} \mathbb{E}\left[ (\widetilde{F}_\gamma - F_\gamma)^2 \right] \lesssim_\gamma \frac{1}{(\alpha^2 n)^{\gamma-1}} + \frac{1}{\alpha^2 n}.$$

For $\gamma > 1$, the rate of the two-step procedure is therefore independent of the alphabet size $K$, unlike the rates of the plug-in and thresholded estimators. Hence, when $K$ is large, this rate is faster than those of the two (non-interactive) estimators. In particular, it is equal to $(\alpha^2 n)^{-1}$ as soon as $\gamma \geq 2$, which is the minimax optimal rate (see next section). Conversely for small $K$ and $\gamma \in (1, 2)$, this rate is slower than those of the plug-in and thresholded estimators. Accordingly, none of the three estimators is overall better than the others, and we discuss the choice of estimator in next section.

## 2.6 Optimality of the Results

Among the three estimators we proposed, the choice of estimators depends on the problem parameters $K, \gamma$. The following recipe leads to a better estimator $\hat{E}_\gamma$. If $K \leq \sqrt{\alpha^2 n}$, define $\hat{E}_\gamma$ as the plug-in estimator $\hat{F}_\gamma$; otherwise (when $K > \sqrt{\alpha^2 n}$), $\hat{E}_\gamma$ is equal to the thresholded estimator $\overline{F}_\gamma$ for $\gamma < 1$, and to the sequentially interactive procedure $\widetilde{F}_\gamma$ for $\gamma > 1$.

**Corollary 2.5.** *The quadratic risk of $\hat{E}_\gamma$ is uniformly bounded over all $p \in \mathcal{P}_K$ by*
1) $\gamma \in (0, 1)$:

$$\sup_{p \in \mathcal{P}_K} \mathbb{E}\left[(\hat{E}_\gamma - F_\gamma)^2\right] \lesssim_\gamma K^{2(1-\gamma)} \wedge \frac{K^2}{(\alpha^2 n)^\gamma} \ . \tag{8}$$

2) $\gamma \in (1, 3/2)$:

$$\sup_{p \in \mathcal{P}_K} \mathbb{E}\left[(\hat{E}_\gamma - F_\gamma)^2\right] \lesssim_\gamma \frac{1}{(\alpha^2 n)^{\gamma-1}} \wedge \left(\frac{K^2}{(\alpha^2 n)^\gamma} + \frac{K^{3-2\gamma}}{\alpha^2 n}\right) \ . \tag{9}$$

3) $\gamma \in (3/2, 2)$:

$$\sup_{p \in \mathcal{P}_K} \mathbb{E}\left[(\hat{E}_\gamma - F_\gamma)^2\right] \lesssim_\gamma \frac{1}{(\alpha^2 n)^{\gamma-1}} \wedge \left(\frac{K^2}{(\alpha^2 n)^\gamma} + \frac{1}{\alpha^2 n}\right) \ . \tag{10}$$

4) $\gamma \geq 2$:

$$\sup_{p \in \mathcal{P}_K} \mathbb{E}\left[(\hat{E}_\gamma - F_\gamma)^2\right] \lesssim_\gamma \frac{1}{\alpha^2 n} \ . \tag{11}$$

The rate (11) for $\gamma \geq 2$ can be seen as the private parametric rate, which is minimax optimal (see lower bounds in Theorem 2.6). Then for $\gamma \in (0, 2)$, the smaller $\gamma$ is, the slower the rates are. Specifically, each upper bound in (8) to (10) is the minimum of two error bounds, with a phase transition at $K \asymp \sqrt{\alpha^2 n}$. Above this transition level, i.e. when $K \gtrsim \sqrt{\alpha^2 n}$, the upper bounds (9) and (10) for $\gamma \in (1, 2)$ are equal to the first error bound $(\alpha^2 n)^{-(\gamma-1)}$ which is free of $K$. Below the transition level, i.e. when $K \lesssim \sqrt{\alpha^2 n}$, they are equal to the second error bound that depends on $K$. Hence, for $\gamma \in (1, 2)$, the upper bounds on the maximal quadratic risk of $\hat{E}_\gamma$ increase with $K$ as long as $K \lesssim \sqrt{\alpha^2 n}$, and become equal to $(\alpha^2 n)^{-(\gamma-1)}$ for any $K$ larger than $\sqrt{\alpha^2 n}$. In contrast, the upper bound (8) for $\gamma \in (0, 1)$ depends on $K$ regardless of the value of $K$. Thus, the rates get slower as the power $\gamma$ decreases, where $\gamma$ can be seen as a smoothness indicator of the function $x \mapsto x^\gamma$ and thus of the functional $F_\gamma = \sum_k p_k^\gamma$.

We give lower bounds over all estimators and all $\alpha$-LDP sequentially interactive mechanisms. Recall that non-interactive mechanisms are just a special case of sequentially interactive mechanisms, thus our lower bounds hold in particular for non-interactive mechanisms. In the special case where $K \lesssim 1$ is a numerical constant, the next lower bounds match the rate of the plug-in estimator (Theorem 2.1) for any $\gamma > 0$, $\gamma \neq 1$.

**Theorem 2.6.** *For any $\gamma > 0$, $\gamma \neq 1$ and integer $K \geq 2$, we have the lower bound*

$$\inf_{Q, \hat{F}} \sup_{p \in \mathcal{P}_K} \mathbb{E}\left[(\hat{F} - F_\gamma)^2\right] \gtrsim_\gamma \frac{1}{\alpha^2 n} + \frac{1}{(\alpha^2 n)^\gamma} \ ,$$

*where the infimum is taken over all estimators $\hat{F}$ and all $\alpha$-LDP sequentially interactive PM $Q$.*

In the general situation of any $K \geq 2$, we thus have the minimax optimal rate $(\alpha^2 n)^{-1}$ when $\gamma \geq 2$, which is achieved by the sequentially interactive estimator (Theorem 2.4). In contrast for $\gamma \in (0, 2)$, the next lower bounds depend on the alphabet size $K$.

**Theorem 2.7.** *For any integer $K \geq 2$, we have the lower bounds*
1) $\gamma \in (0, 1)$:

$$\inf_{Q, \hat{F}} \sup_{p \in \mathcal{P}_K} \mathbb{E}\left[(\hat{F} - F_\gamma)^2\right] \gtrsim_\gamma K^{2(1-\gamma)} \wedge \frac{K^{2-\gamma}}{(n(e^{2\alpha} - e^{-2\alpha})^2)^\gamma} \tag{12}$$

2) $\gamma \in (1, 2)$:

$$\inf_Q \inf_{\hat{F}} \sup_{p \in \mathcal{P}_K} \mathbb{E}\left[(\hat{F} - F_\gamma)^2\right] \gtrsim_\gamma \frac{1}{[(e^{2\alpha} - e^{-2\alpha})^2 n]^{2(\gamma-1)}} \wedge \frac{K^{2-\gamma}}{[(e^{2\alpha} - e^{-2\alpha})^2 n]^\gamma} \tag{13}$$

*where the infimum is taken over all estimators $\hat{F}$ and all $\alpha$-LDP sequentially interactive PM $Q$.*

Note that $e^{2\alpha} - e^{-2\alpha} \asymp \alpha$ when $\alpha$ is small. Then the lower bound (12) for $\gamma \in (0, 1)$ matches, up to a factor $K^\gamma$, the quadratic risk of the thresholded estimator (Theorem 2.3), or equivalently the rate (8) of the combined estimator $\hat{E}_\gamma$. For $\gamma \in (1, 2)$, the situation is slightly more involved, but one can see again that there is a gap of a factor $K^\gamma$ between some terms of the lower bound (13) and the upper bounds (9) and (10) of Corollary 2.5. Besides, reading the proof of lower bounds, one can check that this $K^\gamma$-gap is tightly connected to the gap between the left-terms $1/(\alpha^2 n)^{2(\gamma-1)}$ and $1/(\alpha^2 n)^{\gamma-1}$ of the lower and upper bounds respectively, so that the optimality of the problem actually boils down to reducing this $K^\gamma$-gap.

**Discussion** In this first work on the estimation of the power sum functional $F_\gamma$, we focus on a plug-in estimator based on samples obtained using a non-interactive PM. Although its analogue estimator in the non private case is known to be nearly minimax optimal, we show that our plug-in estimator performs poorly in the LDP setting for large $K$ (proving a tight characterization of its maximal quadratic risk). Hence, practical methods performing well in the non private case should not be systematically transferred to the LDP setting, which requires to design new statistical procedures. We then suggest a correction of this estimator using thresholding, which significantly improves the rates of convergence, removing almost the whole dependence of the risk in the support size $K$ for large $K$. We finally get faster rates by combining these two previous estimators with a sequentially interactive procedure. It is also important to highlight that all privacy mechanisms and estimators introduced here could be further investigated for other functionals, for example the Rényi entropy $H_\gamma$ which is fundamental in information theory and is connected to the power sum $F_\gamma$ via the relation $H_\gamma = \frac{\log F_\gamma}{1-\gamma}$.

We conjecture that our upper bounds are tight, up to some logarithmic factors. In future work, this logarithmic factor should be reduced via the best polynomial approximation method. This is similar to the line of work in the standard setup (without privacy constraint), where the analogue of our plug-in estimator (the MLE) is known to be nearly minimax optimal up to a poly-logarithmic factor [14], while the the best polynomial approximation method closes this logarithmic-gap and is minimax optimal [17, 29]. Leaving aside the logarithmic factors, we conjecture that our lower bounds are not optimal, up to a factor $K^\gamma$. The two fuzzy hypothesis theorem is often used in the standard (non-private) setting to derive lower bounds on the estimation rate of functionals such as the power sum $F_\gamma$. A challenge in the LDP setting is to provide such turnkey tools that help prove universal lower bounds.

Another challenge is to understand when sequentially interactive procedures outperform non-interactive ones. When $\gamma > 1$, we have gained logarithmic factors in our error bounds by considering sequentially interactive procedures. However, because the optimal rate of non-interactive mechanisms is not proven, it is unclear that our logarithmic gap between non-interactive and sequentially interactive actually exists. The estimation of power sum functionals in the context of local differential privacy proves to be a rich topic potentially difficult to solve sharply in all possible cases.

## Acknowledgments and Disclosure of Funding

The authors acknowledge financial support from GENES and the French National Research Agency (ANR), under the grant "Investissements d'Avenir" (LabEx Ecodec/ANR-11-LABX-0047).

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
