# Supplementary Material to 'Locally diferentially private estimation of nonlinear functionals of discrete distributions'

**Cristina Butucea**
CREST, ENSAE, IP Paris
Palaiseau 91120 Cedex, France
`cristina.butucea@ensae.fr`

**Yann Issartel**
CREST, ENSAE, IP Paris
Palaiseau 91120 Cedex, France
`yann.issartel@ensae.fr`

This Supplementary Material contains the proofs of the results in [1] and is consistent in notation with the main paper.

## A Proofs of upper bounds

### A.1 Plug-in estimator

**Proof of 1st bound in Theorem 2.1.** $1°$. *Bias:* We have using the triangle inequality,

$$\left| \mathbb{E}\,\hat{F}_\gamma - F_\gamma \right| = \left| \mathbb{E}\,\hat{F}_\gamma - \sum_{k=1}^{K} p_k^\gamma \right| \leq \sum_{k=1}^{K} \left| \mathbb{E}\,\hat{F}_\gamma(k) - p_k^\gamma \right| \ .$$

Hence, it suffices to upper bound the $k^{\text{th}}$ bias component $|\,\mathbb{E}\,\hat{F}_\gamma(k) - p_k^\gamma|$ for all $k \in [K]$ and $\gamma \neq 1$ (the case $\gamma = 1$ being trivial). We separate the analysis in two different ranges of values of $p_k$. Define $\mathcal{K}_{\geq\tau} = \{k \in [K] : p_k \geq \tau\}$, and $\mathcal{K}_{<\tau} = [K] \setminus \mathcal{K}_{\geq\tau}$. By Lemma B.5 we have

$$\sum_{k \in \mathcal{K}_{<\tau}} \left| \mathbb{E}\,\hat{F}_\gamma(k) - p_k^\gamma \right| \leq C \frac{|\mathcal{K}_{<\tau}|}{(\alpha^2 n)^{\gamma/2}}$$

for a constant $C$ depending only on $\gamma$. Lemma B.7 ensures that

$$\sum_{k \in \mathcal{K}_{\geq\tau}} \left| \mathbb{E}\,\hat{F}_\gamma(k) - p_k^\gamma \right| \leq C' \left( \frac{|\mathcal{K}_{\geq\tau}|}{(\alpha^2 n)^{\gamma/2}} + \mathbb{1}_{\{\gamma \geq 2\}} \frac{\|p^{\geq\tau}\|_{\gamma-2}^{\gamma-2}}{\alpha^2 n} \right)$$

for a constant $C'$ depending only on $\gamma$. Gathering the above inequalities, we have

$$\left| \mathbb{E}\,\hat{F}_\gamma - F_\gamma \right| \leq (C + C') \left( \frac{K}{(\alpha^2 n)^{\gamma/2}} + \mathbb{1}_{\{\gamma \geq 2\}} \frac{\|p^{\geq\tau}\|_{\gamma-2}^{\gamma-2}}{\alpha^2 n} \right). \tag{14}$$

$2°$. *Variance:* By Lemma B.4 , we have $\text{Cov}\big(\hat{F}_\gamma(k), \hat{F}_\gamma(k')\big) \leq 0$ for any $k \neq k' \in [K]$. Hence

$$\text{Var}\left( \sum_{k=1}^{K} \hat{F}_\gamma(k) \right) \leq \sum_{k=1}^{K} \text{Var}\left( \hat{F}_\gamma(k) \right) \ . \tag{15}$$

As in the proof of the bias bound above, we separate our analysis in two different ranges of values of $p_k$. For small $p_k$, we use Lemma B.5 to get

$$\sum_{k \in \mathcal{K}_{<\tau}} \text{Var}\left( \hat{F}_\gamma(k) \right) \leq \widetilde{C} \frac{|\mathcal{K}_{<\tau}|}{(\alpha^2 n)^\gamma} \ ,$$

where $\widetilde{C}$ is a constant depending only on $\gamma$. For large $p_k$, we deduce from Lemma B.8 that

$$\sum_{k \in \mathcal{K}_{\geq \tau}} \mathrm{Var}\left(\hat{F}_\gamma(k)\right) \leq \widetilde{C}'\left(\frac{|\mathcal{K}_{\geq \tau}|}{(\alpha^2 n)^\gamma} + \mathbb{1}_{\{\gamma \geq 1\}} \frac{\|p^{\geq \tau}\|_{2\gamma-2}^{2\gamma-2}}{\alpha^2 n}\right)$$

for a constant $\widetilde{C}'$ depending only on $\gamma$. Then, plugging these bounds into (15), we have

$$\mathrm{Var}\left(\sum_{k=1}^K \hat{F}_\gamma(k)\right) \leq (\widetilde{C} + \widetilde{C}')\left(\frac{K}{(\alpha^2 n)^\gamma} + \mathbb{1}_{\{\gamma \geq 1\}} \frac{\|p^{\geq \tau}\|_{2\gamma-2}^{2\gamma-2}}{\alpha^2 n}\right) \ . \tag{16}$$

The proof of the of 1$^{\text{st}}$ bound in Theorem 2.1 is complete. $\qquad\square$

**Proof of** 2$^{\text{nd}}$ **bound in Theorem 2.1.** We only need to control the second and third terms of the 1$^{\text{st}}$ bound in Theorem 2.1. The squared root of the second term is bounded from above by

$$\frac{\sum_{k=1}^K p_k^{\gamma-2} \mathbb{1}_{\{p_k > \tau\}}}{\alpha^2 n} \leq \sum_{k=1}^K \frac{p_k^{\gamma-1}}{\sqrt{\alpha^2 n}} \frac{p_k^{-1} \mathbb{1}_{\{p_k > \tau\}}}{\sqrt{\alpha^2 n}} \leq \sum_{k=1}^K \frac{p_k^{\gamma-1} c^{-1}}{\sqrt{\alpha^2 n}} = \frac{\|p\|_{\gamma-1}^{\gamma-1} c^{-1}}{\sqrt{\alpha^2 n}} \ .$$

Since $(p_k)_k$ are probabilities, we have $p_k^{\gamma-1} \leq p_k$ for $\gamma \geq 2$ and we can further bound the last display by $\|p\|_{\gamma-1}^{\gamma-1} \leq \sum_{k=1}^K p_k = 1$ for $\gamma \geq 2$. Hence, the second term is bounded by $\mathbb{1}_{\gamma \geq 2}(\alpha^2 n)^{-1}$.

Let us bound the third term. Since $\sum_k p_k = 1$, the number of the significant $p_k \geq \tau$ is necessarily smaller than $\tau^{-1} = c^{-1}\sqrt{\alpha^2 n}$, and thus smaller than $K_{\wedge \tau^{-1}} := K \wedge \sqrt{\alpha^2 n}$. Then, when $\gamma \in (1, 3/2)$, we use the concavity to have $\|p^{\geq \tau}\|_{2\gamma-2}^{2\gamma-2} \leq K_{\wedge \tau^{-1}}^{3-2\gamma}$ for all $p \in \mathcal{P}_K$. When $\gamma \geq 3/2$ we have $\|p^{\geq \tau}\|_{2\gamma-2}^{2\gamma-2} \leq 1$. Therefore, the third term is uniformly bounded over the class $\mathcal{P}_K$ by

$$\mathbb{1}_{\{\gamma \geq 1\}} \frac{\|p^{\geq \tau}\|_{2\gamma-2}^{2\gamma-2}}{\alpha^2 n} \leq \mathbb{1}_{\{\gamma \geq 1\}} \frac{1 \vee K_{\wedge \tau^{-1}}^{3-2\gamma}}{\alpha^2 n} \ .$$

This concludes the proof of the 2$^{\text{nd}}$ bound in Theorem 2.1. $\qquad\square$

## A.2 Thresholded plug-in estimator (proof of Theorem 2.3)

Case $\gamma \in (0, 1)$: Let us check the first bound of Theorem 2.3. We use the concavity of the power function $p^\gamma$ to have $F_\gamma \leq K(\sum_{k=1}^K p_k/K)^\gamma = K^{1-\gamma}$. Then, the quadratic risk of the trivial estimator 0 is bounded by $K^{2(1-\gamma)}$. On the other hand, the quadratic risk of the plug-in $\hat{F}_\gamma$ is bounded by $K^2/(\alpha^2 n)^\gamma$ (Theorem 2.1). Therefore, the quadratic risk of the thresholded estimator $\overline{F}_\gamma := \mathbb{1}_{K \leq \tau^{-1}} \hat{F}_\gamma$ satisfies the first bound of Theorem 2.3.

Case $\gamma > 1$: Recall that $\hat{\tau} \asymp \sqrt{\log(Kn)/(\alpha^2 n)}$. We will prove the next bound on the risk of $\overline{F}_\gamma$,

$$\mathbb{E}\left[(\overline{F}_\gamma - F_\gamma)^2\right] \lesssim_\gamma (K\hat{\tau}^\gamma \wedge \hat{\tau}^{\gamma-1})^2 + \frac{(K \wedge \hat{\tau}^{-1})^{3-2\gamma} \vee 1}{\alpha^2 n} \ . \tag{17}$$

Before that, we check that (17) implies the second inequality of Theorem 2.3.

*(i) Assume that $K \geq \hat{\tau}^{-1}$*, then the RHS of (17) becomes

$$\hat{\tau}^{2(\gamma-1)} + \frac{\hat{\tau}^{2\gamma-3} \vee 1}{\alpha^2 n} \lesssim \frac{(\log(Kn))^{\gamma-1}}{(\alpha^2 n)^{\gamma-1}} + \frac{(\log(Kn))^{\gamma-(3/2)}}{(\alpha^2 n)^{\gamma-(1/2)}} + \frac{1}{\alpha^2 n} \lesssim \frac{(\log(Kn))^{\gamma-1}}{(\alpha^2 n)^{\gamma-1}} + \frac{1}{\alpha^2 n} \ ,$$

where the last inequality follows from the bound

$$\frac{(\log(Kn))^{\gamma-(3/2)}}{(\alpha^2 n)^{\gamma-(1/2)}} \leq \frac{(\log(Kn))^{\gamma-1}}{(\alpha^2 n)^{\gamma-1}} \ ,$$

which is equivalent to $\alpha^2 n \log(Kn) \geq 1$. Hence, (17) is upper bounded by the smallest term of the second inequality of Theorem 2.3.

*(ii) Assume that $K \leq \hat{\tau}^{-1}$, then the RHS of (17) becomes*

$$K^2 \hat{\tau}^{2\gamma} + \frac{K^{3-2\gamma} \vee 1}{\alpha^2 n} \lesssim \frac{K^2 \left(\log(Kn)\right)^\gamma}{(\alpha^2 n)^\gamma} + \frac{1 \vee K^{3-2\gamma}}{\alpha^2 n} \quad,$$

which is the smallest term of the second inequality of Theorem 2.3. Hence, we have proved that the second inequality of Theorem 2.3 follows from (17).

**Proof of** (17). We have the deterministic bound

$$|\overline{F}_\gamma - F_\gamma| \leq \overline{F}_\gamma + F_\gamma \leq K(2^\gamma + 1) \ .$$

Introduce the following event

$$A = \left\{ \exists k \in [K] : \left(\hat{z}_k^{(1)} < \hat{\tau} \text{ and } p_k \geq 3\hat{\tau}/2\right) \text{ or } \left(\hat{z}_k^{(1)} \geq \hat{\tau} \text{ and } p_k < \hat{\tau}/2\right) \right\}$$

and denote the complementary event by $A^c$. We have

$$\mathbb{E}\left[(\overline{F}_\gamma - F_\gamma)^2\right] \leq \mathbb{E}\left[\mathbb{1}_{A^c}(\overline{F}_\gamma - F_\gamma)^2\right] + \mathbb{P}(A)\left(K(2^\gamma + 1)\right)^2 \ . \tag{18}$$

Let us bound the second term of the RHS of (18) by showing that $\mathbb{P}(A) \leq 6/(K^2 n)$. By assumption in the theorem, we have $n \geq 2\log(K)$. This ensures that $n \geq \log(Kn^{1/3})$, which allows us to use Lemma B.3 which gives $\mathbb{P}\left(|\hat{z}_k^{(1)} - p_k| > \hat{\tau}/2\right) \leq 6/(K^3 n)$. Hence, for $p_k \geq 3\hat{\tau}/2$, we have

$$\mathbb{P}\left(\hat{z}_k^{(1)} < \hat{\tau}\right) \leq \frac{6}{K^3 n} \quad,$$

and for $p_k < \hat{\tau}/2$,

$$\mathbb{P}\left(\hat{z}_k^{(1)} \geq \hat{\tau}\right) \leq \frac{6}{K^3 n} \ .$$

We then use the union bound over $k \in [K]$ to get $\mathbb{P}(A) \leq 6/(K^2 n)$. The second term of the RHS of (18) is therefore bounded by $6(2^\gamma + 1)^2/n$.

We now control the first term of the RHS of (18). For any real $a > 0$, we note $\mathcal{K}_{<a} = \{k \in [K] : p_k < a\}$ and $\hat{\mathcal{K}}_{<a} = \{k \in [K] : \hat{z}_k^{(1)} < a\}$, with their respective complementary sets $\mathcal{K}_{\geq a} = [K] \setminus \mathcal{K}_{<a}$ and $\hat{\mathcal{K}}_{\geq a} = [K] \setminus \hat{\mathcal{K}}_{<a}$. Splitting the sum over the $k$ in $\hat{\mathcal{K}}_{<\hat{\tau}}$ and $\hat{\mathcal{K}}_{\geq\hat{\tau}}$ respectively, we get

$$\mathbb{1}_{A^c}(\overline{F}_\gamma - F_\gamma)^2 \leq 2\mathbb{1}_{A^c}\left(\|(p_k)_{k \in \hat{\mathcal{K}}_{<\hat{\tau}}}\|_\gamma^\gamma\right)^2 + 2\mathbb{1}_{A^c}\left(\sum_{k \in \hat{\mathcal{K}}_{\geq\hat{\tau}}} \overline{F}_\gamma(k) - F_\gamma(k)\right)^2 \ .$$

Since $\hat{\mathcal{K}}_{<\hat{\tau}} \subset \mathcal{K}_{<3\hat{\tau}/2}$ on the event $A^c$, we can bound the first term by

$$\mathbb{1}_{A^c}\|(p_k)_{k \in \hat{\mathcal{K}}_{<\hat{\tau}}}\|_\gamma^\gamma \leq \|(p_k)_{k \in \mathcal{K}_{<3\hat{\tau}/2}}\|_\gamma^\gamma \leq K(3\hat{\tau}/2)^\gamma \wedge (3\hat{\tau}/2)^{\gamma-1}$$

for any $\gamma > 1$ and $p \in \mathcal{P}_K$. For the second term, we will use the independence between the data samples $z^{(1)} := (z_1^{(1)}, \ldots, z_n^{(1)})$ and $z^{(2)} := (z_1^{(2)}, \ldots, z_n^{(2)})$. In particular, the set $\hat{\mathcal{K}}_{\geq\hat{\tau}}$ and the event $A^c$ are deterministic conditionally to $z^{(1)}$, so that

$$\mathbb{E}\left[\mathbb{1}_{A^c}\left(\sum_{k \in \hat{\mathcal{K}}_{\geq\hat{\tau}}} \overline{F}_\gamma(k) - F_\gamma(k)\right)^2 \Big| z^{(1)}\right] = \mathbb{1}_{A^c}\mathbb{E}\left[\left(\sum_{k \in \hat{\mathcal{K}}_{\geq\hat{\tau}}} \overline{F}_\gamma(k) - F_\gamma(k)\right)^2 \Big| z^{(1)}\right]$$

$$\leq \mathbb{1}_{A^c} C\left(\frac{|\hat{\mathcal{K}}_{\geq\hat{\tau}}|^2}{(\alpha^2 n)^\gamma} + \frac{|\hat{\mathcal{K}}_{\geq\hat{\tau}}|^{3-2\gamma} \vee 1}{\alpha^2 n}\right)$$

where the last line is similar to the 2$^{\text{nd}}$ bound in Theorem 2.1 with $K$ replaced by $|\hat{\mathcal{K}}_{\geq\hat{\tau}}|$, and where $C$ is some constant depending only on $\gamma$. We can further bound the last display by noting that $\hat{\mathcal{K}}_{\geq\hat{\tau}} \subset \mathcal{K}_{\geq\hat{\tau}/2}$ on the event $A^c$, and $|\mathcal{K}_{\geq\hat{\tau}/2}| \leq K \wedge (\hat{\tau}/2)^{-1}$. Going back to (18), we then have for all $p \in \mathcal{P}_K$,

$$\mathbb{E}\left[(\overline{F}_\gamma - F_\gamma)^2\right] \lesssim_\gamma (K\hat{\tau}^\gamma \wedge \hat{\tau}^{\gamma-1})^2 + \frac{(K \wedge \hat{\tau}^{-1})^2}{(\alpha^2 n)^\gamma} + \frac{(K \wedge \hat{\tau}^{-1})^{3-2\gamma} \vee 1}{\alpha^2 n} + \frac{1}{n}$$

$$\lesssim_\gamma (K\hat{\tau}^\gamma \wedge \hat{\tau}^{\gamma-1})^2 + \frac{(K \wedge \hat{\tau}^{-1})^{3-2\gamma} \vee 1}{\alpha^2 n} \ .$$

The proof of (17) is complete. □

## A.3 Interactive privacy mechanism

**Proof of 1ˢᵗ bound in Theorem 2.4.** 1°. *Bias:* We decompose the expected value of $\widetilde{F}_\gamma$ :

$$\mathbb{E}\,\widetilde{F}_\gamma = \frac{1}{n}\sum_{i=1}^n \mathbb{E}\,\mathbb{E}\left[z_i^{(2)}|z^{(1)}, z^{(2)}\right] = \frac{1}{n}\sum_{i=1}^n \mathbb{E}\,\mathbb{E}\left[\hat{F}_{\gamma-1}^{(1)}(x_i^{(2)})|z^{(1)}, x^{(2)}\right]$$

$$= \sum_{k=1}^K p_k\,\mathbb{E}\,\mathbb{E}\left[\hat{F}_{\gamma-1}^{(1)}(k)|z^{(1)}\right] = \sum_{k=1}^K p_k\,\mathbb{E}\left[\hat{F}_{\gamma-1}^{(1)}(k)\right] \tag{19}$$

so that, for any $\gamma > 1$, $\gamma \neq 2$ (the case $\gamma = 2$ being trivial), we have

$$\left|\mathbb{E}\,\widetilde{F}_\gamma - \sum_{k=1}^K p_k^\gamma\right| \leq \sum_{k=1}^K p_k\left|\mathbb{E}\,\hat{F}_{\gamma-1}^{(1)}(k) - p_k^{\gamma-1}\right|$$

$$\leq C\left(\frac{1}{(\alpha^2 n)^{(\gamma-1)/2}} + \mathbb{1}_{\{\gamma \geq 3\}}\frac{\|p^{\geq\tau}\|_{\gamma-2}^{\gamma-2}}{\alpha^2 n}\right) \tag{20}$$

using Lemma B.5 and B.7 and $\sum_k p_k = 1$, where $C$ is a constant depending only on $\gamma$.

2°. *Variance:* By the law of total variance we have

$$\mathrm{Var}\left(\widetilde{F}_\gamma\right) = \mathbb{E}\left[\mathrm{Var}\left(\widetilde{F}_\gamma|z^{(1)}\right)\right] + \mathrm{Var}\left(\mathbb{E}\left[\widetilde{F}_\gamma|z^{(1)}\right]\right)\ . \tag{21}$$

We control the first term in the RHS of (21):

$$\mathrm{Var}\left(\widetilde{F}_\gamma|z^{(1)}\right) = \frac{1}{n}\mathrm{Var}\left(z_1^{(2)}|z^{(1)}\right) \leq \frac{1}{n}\mathbb{E}\left[\left(z_1^{(2)}\right)^2|z^{(1)}\right]$$

$$= \frac{2^{2\gamma-1}}{n}\left(\frac{e^\alpha + 1}{e^\alpha - 1}\right)^2 \leq \frac{2^{2\gamma+1}}{\alpha^2 n}$$

where we used $(\frac{e^\alpha+1}{e^\alpha-1})^2 = (1 + \frac{1}{e^\alpha-1})^2 \leq (1 + \frac{1}{\alpha})^2 \leq \frac{4}{\alpha^2}$. For the second term in the RHS of (21), we have using (19)

$$\mathrm{Var}\left(\mathbb{E}\left[\widetilde{F}_\gamma|Z^{(1)}\right]\right) = \mathrm{Var}\left(\sum_{k=1}^K p_k\hat{F}_{\gamma-1}^{(1)}(k)\right) \leq \sum_{k=1}^K p_k^2\mathrm{Var}\left(\hat{F}_{\gamma-1}^{(1)}(k)\right)$$

where the inequality can be deduced from Lemma B.4. Then, by Lemma B.5 and B.8,

$$\sum_{k=1}^K p_k^2\mathrm{Var}\left(\hat{F}_{\gamma-1}^{(1)}(k)\right) \leq \widetilde{C}\left(\frac{\|p\|_2^2}{(\alpha^2 n)^{\gamma-1}} + \mathbb{1}_{\{\gamma \geq 2\}}\frac{\|p^{\geq\tau}\|_{2\gamma-2}^{2\gamma-2}}{\alpha^2 n}\right)$$

for a constant $\widetilde{C}$ depending only $\gamma$. The proof of the 1ˢᵗ bound in Theorem 2.4 is complete. □

**Proof of 2ⁿᵈ bound in Theorem 2.4.** The desired bound follows from the 1ˢᵗ bound of Theorem 2.4 and the fact that $\mathbb{1}_{\{\gamma \geq 3\}}\|p^{\geq\tau}\|_{\gamma-2}^{\gamma-2} \leq 1$ and $\mathbb{1}_{\{\gamma \geq 2\}}\|p^{\geq\tau}\|_{2\gamma-2}^{2\gamma-2} \leq \|p\|_2^2 \leq 1$ for all $p \in \mathcal{P}_K$. □

# B  Main lemmas for upper bounds

We use the notations $\hat{x}_k = \frac{1}{n}\sum_{i=1}^n \mathbb{1}_{\{x_i=k\}}$ and $\hat{w}_k = \frac{1}{n}\sum_{i=1}^n w_{ik}$, so that $\hat{z}_k = \hat{x}_k + \frac{\sigma}{\alpha}\hat{w}_k$. We consider $\alpha \in (0, \infty)$ in this Appendix B, unlike in the main section of the paper where we assumed that $\alpha \in (0, 1)$ and $\alpha^2 n \geq 1$.

## B.1  Concentration of $\hat{z}_k$

We control the concentration of $\hat{z}_k$ in the next lemma.

**Lemma B.1.** *For any $\alpha \in (0, \infty)$ and any $r > 0$, we have*

$$\mathbb{E}\left[|\hat{z}_k - p_k|^r\right] \leq \frac{C_{BL,r}}{((\alpha^2 \wedge 1)n)^{r/2}} \ ,$$

$$\mathbb{E}\left[|\hat{z}_k|^r\right] \leq \frac{2^r C_{BL,r}}{((\alpha^2 \wedge 1)n)^{r/2}} + 2^r p_k^r \ ,$$

*where $C_{BL,r}$ is a constant depending only on $r$. Besides,*

$$\mathbb{P}(\hat{z}_k < \frac{p_k}{2}) \leq 3 \exp\left[-\frac{n}{128}\left(\frac{(\alpha \wedge 1)p_k}{\sigma}\right)^2\right] \ .$$

**Proof of Lemma B.1.** By (35) in Lemma C.1 and (37) in Lemma C.2, we have for any $r > 0$,

$$\mathbb{E}\left[|\hat{z}_k - p_k|^r\right] \leq 2^r \mathbb{E}\left[|\hat{x}_k - p_k|^r\right] + 2^r \mathbb{E}\left[\left(\frac{\sigma|\hat{w}_k|}{\alpha}\right)^r\right] \leq \frac{2^r C_{B,r}}{n^{r/2}} + \frac{(2\sigma)^r C_{L,r}}{(\alpha^2 n)^{r/2}}$$

$$\leq \frac{2^r \left(C_{B,r} + \sigma^r C_{L,r}\right)}{((\alpha^2 \wedge 1)n)^{r/2}}$$

where $C_{B,r}$ and $C_{L,r}$ are constants that only depend on $r$. Then, denoting $C_{BL,r} = 2^r \left(C_{B,r} + \sigma^r C_{L,r}\right)$, we have

$$\mathbb{E}\left[|\hat{z}_k|^r\right] = \mathbb{E}\left[|\hat{z}_k - p_k + p_k|^r\right] \leq 2^r \mathbb{E}\left[|\hat{z}_k - p_k|^r\right] + 2^r p_k^r$$

$$\leq \frac{2^r C_{BL,r}}{((\alpha^2 \wedge 1)n)^{r/2}} + 2^r p_k^r \ .$$

Finally, by (32) in Lemma C.1 and (36) in Lemma C.2, we have

$$\mathbb{P}(\hat{z}_k < \frac{p_k}{2}) \leq \mathbb{P}(\hat{x}_k < \frac{3p_k}{4}) + \mathbb{P}(\frac{\sigma\hat{w}_k}{\alpha} < -\frac{p_k}{4}) \leq e^{-\left(\frac{1}{4}\right)^2 \frac{np_k}{2}} + e^{-\frac{n}{8}\left(\frac{\alpha p_k}{4\sigma}\right)^2} + e^{-\frac{n}{4}\left(\frac{\alpha p_k}{4\sigma}\right)}$$

$$\leq 3\,e^{-\frac{n}{128\sigma^2}((\alpha\wedge 1)p_k)^2} \ .$$

The proof of Lemma B.1 is complete. $\qquad\qquad\square$

Recall that $\hat{F}_\gamma(k) = \left(T_{[0,2]}[\hat{z}_k]\right)^\gamma$. We bound the difference between the expectations of $T_{[0,2]}[\hat{z}_k]$ and $\hat{z}_k$ in the next lemma.

**Lemma B.2.** *We have for any $\alpha \in (0, \infty)$,*

$$\left|\mathbb{E}\left[T_{[0,2]}[\hat{z}_k]\right] - p_k\right| \leq \frac{2p_k^{-1}}{(\alpha^2 \wedge 1)n}\left(\sigma^2 C_{L,2} + \frac{16\gamma}{e}\right) \ .$$

**Proof of Lemma B.2.** Recall that $\hat{z}_k = \hat{x}_k + \frac{\sigma}{\alpha}\hat{w}_k$, and define $\epsilon_k$ by $T_{[0,2]}[\hat{z}_k] = \hat{x}_k + \epsilon_k$. Then $\mathbb{E}\left[T_{[0,2]}[\hat{z}_k]\right] - p_k = \mathbb{E}[\epsilon_k]$ and it suffices to bound $|\mathbb{E}[\epsilon_k]|$. Introducing the event $A = \{|\frac{\sigma}{\alpha}\hat{w}_k| < \hat{x}_k\}$ and the complementary event $A^c$, we note first that $A \subseteq \{\hat{z}_k \in [0, 2]\}$ and thus $\epsilon_k = \frac{\sigma}{\alpha}\hat{w}_k$ on $A$. We have

$$|\mathbb{E}[\epsilon_k]| \leq |\mathbb{E}[\epsilon_k \mathbb{1}_A]| + |\mathbb{E}[\epsilon_k \mathbb{1}_{A^c}]| = |\mathbb{E}\left[\frac{\sigma}{\alpha}\hat{w}_k \mathbb{1}_A\right]| + |\mathbb{E}[\epsilon_k \mathbb{1}_{A^c}]|$$

$$= |\mathbb{E}\,\mathbb{E}\left[\frac{\sigma}{\alpha}\hat{w}_k \mathbb{1}_A \Big| \hat{x}_k\right]| + |\mathbb{E}[\epsilon_k \mathbb{1}_{A^c}]|$$

$$= |\mathbb{E}[\epsilon_k \mathbb{1}_{A^c}]|$$

since $\hat{w}_k$ is a centered and symmetric random variable that is independent of $\hat{x}_k$. Using the event $B = \{2p_k \geq \hat{x}_k \geq p_k/2\}$ and the complementary event $B^c$, we have

$$|\mathbb{E}[\epsilon_k \mathbb{1}_{A^c}]| \leq \mathbb{E}\left[|\epsilon_k|\mathbb{1}_{A^c \cap B}\right] + \mathbb{E}\left[|\epsilon_k|\mathbb{1}_{A^c \cap B^c}\right] \leq \mathbb{E}\left[|\epsilon_k|\mathbb{1}_{\{\frac{\sigma}{\alpha}|\hat{w}_k| \geq \frac{1}{2}p_k\}}\right] + 2\,\mathbb{E}[\mathbb{1}_{B^c}]$$

$$\leq \mathbb{E}\left[\frac{\sigma}{\alpha}|\hat{w}_k|\mathbb{1}_{\{\frac{\sigma}{\alpha}|\hat{w}_k| \geq \frac{1}{2}p_k\}}\right] + 4e^{-\frac{1}{8}np_k}$$

$$= 2p_k^{-1}\left(\mathbb{E}\left[\frac{p_k}{2}|\frac{\sigma}{\alpha}\hat{w}_k|\mathbb{1}_{\{\frac{\sigma}{\alpha}|\hat{w}_k| \geq \frac{1}{2}p_k\}}\right] + 2p_k e^{-\frac{1}{8}np_k}\right)$$

$$\leq 2p_k^{-1}\left(\mathbb{E}\left[|\frac{\sigma}{\alpha}\hat{w}_k|^2\right] + 2p_k e^{-\frac{1}{8}np_k}\right)$$

where we invoked (32-33) from Lemma C.1 in the second line. Then, by (37) from Lemma C.2,

$$\left| \mathbb{E}\left[ \epsilon_k \mathbb{1}_{A^c} \right] \right| \le 2p_k^{-1}\left( \frac{\sigma^2 C_{L,2}}{\alpha^2 n} + 2p_k e^{-np_k/8} \right) \le 2p_k^{-1}\left( \frac{\sigma^2 C_{L,2}}{\alpha^2 n} + \frac{16\gamma}{en} \right)$$

where we used $xe^{-cnx} \le \frac{\gamma}{cen}$ for any $x \in [0,1]$ and any $c > 0$. This concludes the proof of Lemma B.2. $\qquad\square$

**Lemma B.3.** *For any $\alpha \in (0,\infty)$, and integers $K, n$ satisfying $n \ge \log(Kn^{1/3})$, we have*

$$\mathbb{P}\left( |\hat{z}_k - p_k| > 96\sigma\sqrt{\frac{\log(Kn^{1/3})}{(\alpha^2 \wedge 1)n}} \right) \le \frac{6}{K^3 n} \ .$$

**Proof of Lemma B.3.** Denoting $\delta = c_1\sigma\sqrt{\frac{\log(Kn^{1/3})}{(\alpha^2\wedge 1)n}}$ with $c_1 \ge 1$ a numerical constant to be set later, we get from (34) in Lemma C.1 and (36) in Lemma C.2 that

$$\mathbb{P}(|\hat{z}_k - p_k| > \delta) \le \mathbb{P}(|\hat{x}_k - p_k| > \frac{\delta}{2}) + \mathbb{P}(\frac{\sigma|\hat{w}_k|}{\alpha} > \frac{\delta}{2}) \le 2\left( e^{-\frac{n\delta^2}{2}} + e^{-\frac{n(\alpha\delta/\sigma)^2}{32}} + e^{-\frac{n(\alpha\delta/\sigma)}{8}} \right)$$

$$\le 6\, e^{-\frac{c_1 \log(Kn^{1/3})}{32}}$$

which is upper bounded by $6/(K^3 n)$ for $c_1 = 96$. Lemma B.3 is proved. $\qquad\square$

**Lemma B.4.** *We have* $\ \mathrm{Cov}\big(\hat{F}_\gamma(k), \hat{F}_\gamma(k')\big) \le 0$ *for any $k, k' \in [K]$, $k \ne k'$, and any $\gamma > 0$.*

**Proof of Lemma B.4.** We first state the definition of the negative association property.

*Definition* (See [5]) Random variables $u_1,\ldots,u_K$ are said to be negatively associated (NA) if for every pair of disjoint subsets $A_1, A_2$ of $\{1,\ldots,K\}$, and any component-wise increasing functions $f_1, f_2$,

$$\mathrm{Cov}\big(f_1(u_i, i \in A_1), f_2(u_j, j \in A_2)\big) \le 0 \ . \tag{22}$$

By corollary 5 of Jiao et al. [4], random variables that are drawn from a multinomial distribution, are NA. Hence, the random variables $\hat{X} = (\hat{x}_1,\ldots,\hat{x}_K)$ are NA since $(\hat{x}_1,\ldots,\hat{x}_K)$ follows a multinomial distribution $\sim \mathcal{M}(n; (p_k)_{k\in[K]})$. Besides, the $\hat{W} = (\hat{w}_k)_{k\in[K]}$ are NA, as any set of independent random variables are NA [5]. Then, we get that $(\hat{X}, \hat{W}) = (\hat{x}_1,\ldots,\hat{x}_K, \hat{w}_1,\ldots,\hat{w}_K)$ are NA since a standard closure property of NA is that the union of two independent sets of NA random variables is NA [5]. We can therefore use the definition (22) of NA random variables to have

$$\mathrm{Cov}\big(f_k(\hat{X},\hat{W}), f_{k'}(\hat{X},\hat{W})\big) \le 0 \ , \qquad \forall k,k' \in [K], k \ne k'$$

for $f_k[(\hat{x}_1,\ldots,\hat{x}_K,\hat{w}_1,\ldots,\hat{w}_K)] = \big[T_{[0,2]}(\hat{x}_k + \sigma\hat{w}_k/\alpha)\big]^\gamma$, which are component-wise increasing functions. The proof of Lemma B.4 is complete. $\qquad\square$

## B.2 Bias and Variance on small values of $p_k$

**Lemma B.5.** *Let $\gamma, \alpha \in (0,\infty)$ and $k \in [K]$ and $c > 1$ be any numerical constant. If $p_k \le c/\sqrt{(\alpha^2 \wedge 1)n}$, then*

$$\left| \mathbb{E}\,\hat{F}_\gamma(k) - p_k^\gamma \right| \le \frac{C}{((\alpha^2 \wedge 1)n)^{\gamma/2}} \ ,$$

$$\mathrm{Var}\left( \hat{F}_\gamma(k) \right) \le \frac{C'}{((\alpha^2 \wedge 1)n)^\gamma} \ ,$$

*where $C, C'$ are constants depending only on $\gamma$ and $c$.*

**Proof of Lemma B.5.** Recall that $\hat{F}_\gamma(k) = \left(T_{[0,2]}\left[\hat{z}_k\right]\right)^\gamma$. We have for any $s = 1, 2$,

$$\mathbb{E}\left[(\hat{F}_\gamma(k))^s\right] = \mathbb{E}\left[\left(T_{[0,2]}\left[\hat{z}_k\right]\right)^{s\gamma}\right] \leq \mathbb{E}\left[|\hat{z}_k|^{s\gamma}\right] \leq \frac{2^{s\gamma}C_{BL,s\gamma}}{((\alpha^2 \wedge 1)n)^{s\gamma/2}} + 2^{s\gamma}p_k^{s\gamma}$$

using Lemma B.1. Then, we take $s = 1$ to obtain the first bound announced in the lemma:

$$\left|\mathbb{E}\left[\hat{F}_\gamma(k)\right] - p_k^\gamma\right| \leq \mathbb{E}\left[\hat{F}_\gamma(k)\right] + p_k^\gamma \leq \frac{2^\gamma C_{BL,\gamma}}{((\alpha^2 \wedge 1)n)^{\gamma/2}} + (2^\gamma + 1)p_k^\gamma$$

$$\leq \frac{2^\gamma C_{BL,\gamma} + (2^\gamma + 1)c^\gamma}{((\alpha^2 \wedge 1)n)^{\gamma/2}}$$

since $p_k \leq c/\sqrt{(\alpha^2 \wedge 1)n}$. We finally take $s = 2$ to get the second bound of the lemma:

$$\mathrm{Var}\left(\hat{F}_\gamma(k)\right) \leq \mathbb{E}\left[\hat{F}_\gamma(k)^2\right] \leq \frac{2^{2\gamma}C_{BL,2\gamma} + 2^{2\gamma}c^{2\gamma}}{((\alpha^2 \wedge 1)n)^\gamma} .$$

Lemma B.5 is proved. $\qquad\qquad\square$

## B.3  Bias and Variance on large values of $p_k$

**Lemma B.6.** *For any $\gamma, \alpha \in (0, \infty)$ and $k \in [K]$ with $p_k \in (0, 1]$, we have*

$$\left|\mathbb{E}\left[\hat{F}_\gamma(k)^s\right] - p_k^{s\gamma}\right| \leq C\left(p_k^{s\gamma}e^{-\frac{n}{128\sigma^2}((\alpha\wedge 1)p_k)^2} + \frac{\mathbb{1}_{\{s\gamma\geq 2\}}}{((\alpha^2 \wedge 1)n)^{s\gamma/2}} + \frac{p_k^{s\gamma-2}}{(\alpha^2 \wedge 1)n}\right), \quad \forall s = 1, 2,$$

*where $C$ is a constant depending only on $\gamma$.*

The proof of Lemma B.6 is inspired by the variance bound [4, Lemma 28] as it is based on Taylor's formula with the second derivatives of $x^\gamma$ and $x^{2\gamma}$. However, the result in [4] holds for $\gamma \in (0, 1)$ in the case of direct observations (no privacy), whereas Lemma B.6 holds for any $\gamma > 0$ in the case of sanitized observations (privacy). We postpone the (relatively long) proof to the end of section B.3.

**Lemma B.7.** *Let $\gamma, \alpha \in (0, \infty)$ and $k \in [K]$ and $c > 0$ be any numerical constant. If $p_k \geq c/\sqrt{(\alpha^2 \wedge 1)n}$, then*

$$\left|\mathbb{E}\left[\hat{F}_\gamma(k)^s\right] - p_k^{s\gamma}\right| \leq C\left(\frac{1}{((\alpha^2 \wedge 1)n)^{s\gamma/2}} + \mathbb{1}_{\{s\gamma\geq 2\}}\frac{p_k^{s\gamma-2}}{(\alpha^2 \wedge 1)n}\right), \quad \forall s = 1, 2,$$

*where $C$ is a constant depending only on $\gamma$ and $c$.*

**Proof of Lemma B.7.** We invoke Lemma B.6. We bound the first error term

$$p_k^{s\gamma}e^{-\frac{n}{128\sigma^2}((\alpha\wedge 1)p_k)^2} \leq \left(\frac{64\sigma^2 s\gamma}{(\alpha^2 \wedge 1)en}\right)^{s\gamma/2}$$

where we used $x^{s\gamma}e^{-cnx^2} \leq \left(\frac{s\gamma}{2cen}\right)^{s\gamma/2}$ for $x \in [0, 1]$ and any $c > 0$. The third error term of Lemma B.6 satisfies, for $s\gamma \in (0, 2)$

$$\mathbb{1}_{\{s\gamma\in(0,2)\}}\frac{p_k^{s\gamma-2}}{(\alpha^2 \wedge 1)n} \leq \frac{\left(\frac{c}{\sqrt{(\alpha^2\wedge 1)n}}\right)^{s\gamma-2}}{(\alpha^2 \wedge 1)n} \leq \frac{c^{s\gamma-2}}{((\alpha^2 \wedge 1)n)^{s\gamma/2}} \tag{23}$$

since $p_k \geq c/\sqrt{(\alpha^2 \wedge 1)n}$. The proof of Lemma B.7 is complete. $\qquad\square$

**Lemma B.8.** *Under the assumptions of Lemma B.7, we have*

$$\mathrm{Var}\left(\hat{F}_\gamma(k)\right) \leq C\left(\frac{1}{((\alpha^2 \wedge 1)n)^\gamma} + \mathbb{1}_{\{\gamma\geq 1\}}\frac{p_k^{2\gamma-2}}{(\alpha^2 \wedge 1)n}\right)$$

*for a constant $C$ depending only on $\gamma$ (and $c$).*

**Proof of Lemma B.8.** We have, similarly to [4],

$$
\begin{aligned}
\mathrm{Var}\left(\hat{F}_\gamma(k)\right) &= \mathbb{E}\left[\hat{F}_\gamma(k)^2\right] - \left(\mathbb{E}\,\hat{F}_\gamma(k)\right)^2 = \mathbb{E}\left[\hat{F}_\gamma(k)^2\right] - p_k^{2\gamma} + p_k^{2\gamma} - \left(\mathbb{E}\,\hat{F}_\gamma(k)\right)^2 \\
&\le \left|\mathbb{E}\left[\hat{F}_\gamma(k)^2\right] - p_k^{2\gamma}\right| + \left|p_k^{2\gamma} - \left(\mathbb{E}\,\hat{F}_\gamma(k) - p_k^\gamma + p_k^\gamma\right)^2\right| \\
&\le \left|\mathbb{E}\left[\hat{F}_\gamma(k)^2\right] - p_k^{2\gamma}\right| + \left|\mathbb{E}\,\hat{F}_\gamma(k) - p_k^\gamma\right|^2 + 2p_k^\gamma\left|\mathbb{E}\,\hat{F}_\gamma(k) - p_k^\gamma\right| .
\end{aligned} \tag{24}
$$

Using Lemma B.7 to bound the two first terms of (24), and Lemma B.6 for the last term, we get

$$
\begin{aligned}
\mathrm{Var}\left(\hat{F}_\gamma(k)\right) \le C\bigg( & \frac{1}{((\alpha^2 \wedge 1)n)^\gamma} + \mathbb{1}_{\{\gamma \ge 1\}}\frac{p_k^{2\gamma-2}}{(\alpha^2 \wedge 1)n} \\
& + \frac{1}{((\alpha^2 \wedge 1)n)^\gamma} + \mathbb{1}_{\{\gamma \ge 2\}}\frac{p_k^{2(\gamma-2)}}{((\alpha^2 \wedge 1)n)^2} \\
& + 2p_k^{2\gamma}e^{-\frac{n}{128\sigma^2}((\alpha\wedge 1)p_k)^2} + \frac{2p_k^\gamma\mathbb{1}_{\{\gamma \ge 2\}}}{((\alpha^2 \wedge 1)n)^{\gamma/2}} + \frac{2p_k^{2\gamma-2}}{(\alpha^2 \wedge 1)n}\bigg).
\end{aligned} \tag{25}
$$

We bound the fifth term of (25):

$$
2p_k^{2\gamma}e^{-\frac{n}{128\sigma^2}((\alpha\wedge 1)p_k)^2} \le 2\left(\frac{128\sigma^2\gamma}{(\alpha^2 \wedge 1)en}\right)^\gamma
$$

using $x^{2\gamma}e^{-c'nx^2} \le \left(\frac{\gamma}{c'en}\right)^\gamma$ for any $x \in [0,1]$ and any $c' > 0$. Hence, the first, third and fifth terms of (25) are of the order of $((\alpha^2 \wedge 1)n)^{-\gamma}$ at most. We now bound the fourth term of (25) using $p_k \ge c/\sqrt{(\alpha^2 \wedge 1)n}$ :

$$
\frac{p_k^{2(\gamma-2)}}{((\alpha^2 \wedge 1)n)^2} = \frac{p_k^{2\gamma-2}p_k^{-2}}{((\alpha^2 \wedge 1)n)^2} \le \frac{p_k^{2\gamma-2}}{c^2(\alpha^2 \wedge 1)n}
$$

and similarly the sixth term of (25):

$$
\frac{2p_k^\gamma\mathbb{1}_{\{\gamma \ge 2\}}}{((\alpha^2 \wedge 1)n)^{1+(\gamma/2)-1}} \le \frac{2p_k^\gamma(p_k/c)^{\gamma-2}\mathbb{1}_{\{\gamma \ge 2\}}}{(\alpha^2 \wedge 1)n} = \frac{2p_k^{2\gamma-2}\mathbb{1}_{\{\gamma \ge 2\}}}{c^{\gamma-2}(\alpha^2 \wedge 1)n} .
$$

Hence, we have the desired bound for the second, fourth and sixth terms of (25). Finally, for the last term of (25) we have

$$
\frac{p_k^{2\gamma-2}}{(\alpha^2 \wedge 1)n} = \frac{p_k^{2\gamma-2}\mathbb{1}_{\{\gamma \in (0,1)\}}}{(\alpha^2 \wedge 1)n} + \frac{p_k^{2\gamma-2}\mathbb{1}_{\{\gamma \ge 1\}}}{(\alpha^2 \wedge 1)n} \le \frac{2c^{2\gamma-2}}{((\alpha^2 \wedge 1)n)^\gamma} + \frac{p_k^{2\gamma-2}\mathbb{1}_{\{\gamma \ge 1\}}}{(\alpha^2 \wedge 1)n}
$$

using (23) for $s = 2$. This concludes the proof of of Lemma B.8. $\qquad\square$

**Proof of Lemma B.6.** Denoting $f_s(x) = x^{s\gamma}$ for $s = 1, 2$, and $Y = T_{[0,2]}[\hat{z}_k]$, we have by Taylor's formula,

$$
f_s(Y) = f_s(p_k) + f_s'(p_k)(Y - p_k) + R(Y, p_k) \tag{26}
$$

where the remainder is defined by

$$
R(Y, p_k) = \int_{p_k}^Y (Y - w)f_s''(w)dw = \frac{1}{2}f_s''(w_Y)(Y - p_k)^2 \tag{27}
$$

where $w_Y$ lies between $Y$ and $p_k$. We get

$$
|\mathbb{E}\,f_s(Y) - f_s(p_k)| \le |\mathbb{E}\,R(Y, p_k)| + |\mathbb{E}\,f_s'(p_k)(Y - p_k)| . \tag{28}
$$

Thus, to prove the lemma, it suffices to bound the remainder $|\mathbb{E}\,R(Y, p_k)|$ and the first order term $|\mathbb{E}\,f_s'(p_k)(Y - p_k)|$. We control the latter using Lemma B.2,

$$
|\mathbb{E}\,f_s'(p_k)(Y - p_k)| = s\gamma p_k^{s\gamma-1}|\mathbb{E}(Y - p_k)| \le \frac{2s\gamma p^{s\gamma-2}}{(\alpha^2 \wedge 1)n}\left(\sigma^2 C_{L,2} + \frac{16\gamma}{e}\right) .
$$

For the remainder, we use the decomposition

$$|\mathbb{E}\,R(Y,p_k)| \le \mathbb{E}\left[|R(Y,p_k)|\mathbb{1}(Y < p_k/2)\right] + \mathbb{E}\left[|R(Y,p_k)|\mathbb{1}(Y \ge p_k/2)\right] \qquad (29)$$

and we bound separately the two terms of the RHS.

$1°$. *First term in the RHS of* (29).

$$
\begin{aligned}
\mathbb{E}\left[|R(Y,p_k)|\mathbb{1}(Y < p_k/2)\right] &\le \sup_{y \le p_k/2}|R(y,p_k)|\,\mathbb{E}\left[\mathbb{1}(Y < p_k/2)\right] \\
&= \sup_{y \le p_k/2}|R(y,p_k)|\,\mathbb{E}\left[\mathbb{1}(\hat{z}_k < p_k/2)\right] \\
&\le \sup_{y \le p_k/2}|R(y,p_k)|\,3\,e^{-\frac{n}{128\sigma^2}((\alpha\wedge 1)p_k)^2}
\end{aligned}
$$

using Lemma B.1. We control $R(y,p_k)$ for any $y \in [0, p_k/2]$,

$$
\begin{aligned}
|R(y,p_k)| &\le \int_y^{p_k}(w-y)|f_s''(w)|dw \le \int_y^{p_k}(w-y)s\gamma|s\gamma-1|w^{s\gamma-2}dw \\
&\le s\gamma|s\gamma-1|\int_y^{p_k}w^{s\gamma-1}dw \le s\gamma|s\gamma-1|\int_0^{p_k}w^{s\gamma-1}dw = |s\gamma-1|p_k^{s\gamma} \ .
\end{aligned}
$$

We gather the last two displays to get

$$\mathbb{E}\left[|R(Y,p_k)|\mathbb{1}(Y < p_k/2)\right] \le |s\gamma-1|p_k^{s\gamma}\,3\,e^{-\frac{n}{128\sigma^2}((\alpha\wedge 1)p_k)^2} \ .$$

$2°$. *Second term in the RHS of* (29). We separate our analysis in two different ranges of values of $\gamma$.

$2°.1$. *Case* $s\gamma \in (0,2)$: Starting from (27) we have

$$
\begin{aligned}
\mathbb{E}\left[|R(Y,p_k)|\mathbb{1}(Y \ge p_k/2)\right] &= \frac{s\gamma|s\gamma-1|}{2}\,\mathbb{E}\left[w_Y^{s\gamma-2}(Y-p_k)^2\mathbb{1}(Y \ge p_k/2)\right] \qquad (30) \\
&\le \frac{s\gamma|s\gamma-1|}{2}\left(\frac{p_k}{2}\right)^{s\gamma-2}\mathbb{E}\left[(Y-p_k)^2\right] \\
&\le s\gamma|s\gamma-1|2^{1-s\gamma}p_k^{s\gamma-2}\frac{C_{BL,2}}{(\alpha^2\wedge 1)n}
\end{aligned}
$$

where we used $\mathbb{E}\left[(Y-p_k)^2\right] \le \mathbb{E}\left[(\hat{z}_k-p_k)^2\right]$ and Lemma B.1.

$2°.2$. *Case* $s\gamma \ge 2$: A plug of $w_Y^{s\gamma-2} \le p_k^{s\gamma-2} + Y^{s\gamma-2}$ into (30) gives

$$\mathbb{E}\left[|R(Y,p_k)|\mathbb{1}(Y \ge p_k/2)\right] \le \frac{s\gamma|s\gamma-1|}{2}\,\mathbb{E}\left[(p_k^{s\gamma-2}+Y^{s\gamma-2})(Y-p_k)^2\mathbb{1}(Y \ge p_k/2)\right]. \quad (31)$$

We bound the first part of (31) as in (30),

$$\mathbb{E}\left[p_k^{s\gamma-2}(Y-p_k)^2\mathbb{1}(Y \ge p_k/2)\right] \le p_k^{s\gamma-2}\frac{C_{BL,2}}{((\alpha^2\wedge 1)n)} \ .$$

For the second part of (31), we get from Cauchy-Schwarz that

$$
\begin{aligned}
\mathbb{E}\left[Y^{s\gamma-2}(Y-p_k)^2\mathbb{1}(Y \ge 2p_k)\right] &\le \mathbb{E}\left[Y^{2(s\gamma-2)}\right]^{1/2}\mathbb{E}\left[(Y-p_k)^4\right]^{1/2} \\
&\le \left(\frac{2^{2(s\gamma-2)}C_{BL,2(s\gamma-2)}}{((\alpha^2\wedge 1)n)^{s\gamma-2}} + 2^{2(s\gamma-2)}p_k^{2(s\gamma-2)}\right)^{1/2}\left(\frac{C_{BL,4}}{((\alpha^2\wedge 1)n)^2}\right)^{1/2} \\
&\le \left(\frac{2^{s\gamma-2}\sqrt{C_{BL,2(s\gamma-2)}}}{((\alpha^2\wedge 1)n)^{(s\gamma-2)/2}} + 2^{s\gamma-2}p_k^{s\gamma-2}\right)\frac{\sqrt{C_{BL,4}}}{(\alpha^2\wedge 1)n}
\end{aligned}
$$

where in the second inequality we used $\mathbb{E}\left[Y^{2r}\right] \le \mathbb{E}\left[\hat{z}_k^{2r}\right]$ and $\mathbb{E}\left[(Y-p_k)^{2r}\right] \le \mathbb{E}\left[(\hat{z}_k-p_k)^{2r}\right]$ for any $r > 0$ and Lemma B.1; in the third inequality we used $\sqrt{a+b} \le \sqrt{a}+\sqrt{b}$ for any $a,b > 0$. A plug of the last two displays into (31) concludes the case $s\gamma \ge 2$.

Going back to (29), we have bounded the remainder $\mathbb{E}\,R(Y,p_k)$. Lemma B.6 is proved. $\qquad \square$

# C Auxiliary lemmas for upper bounds

**Lemma C.1.** *Let $p \in (0, 1]$, and $x_1, \ldots, x_n \overset{iid}{\sim} \mathrm{B}(p)$ be independent Bernoulli random variables with parameter $p$. Then, the mean $\hat{x} = \frac{1}{n} \sum_{i=1}^n x_i$ satisfies, for any $\delta > 0$,*

$$\mathbb{P}\left(\hat{x} \le (1 - \delta)p\right) \le e^{-\frac{\delta^2 np}{2}} \quad, \tag{32}$$

$$\mathbb{P}\left(\hat{x} \ge (1 + \delta)p\right) \le e^{-\frac{\delta^2 np}{2+\delta}} \quad, \tag{33}$$

*and*

$$\mathbb{P}(|\hat{x} - p| \ge \delta) \le 2e^{-2\delta^2 n} \quad. \tag{34}$$

*We also have, for any $r > 0$,*

$$\mathbb{E}\left[|\hat{x} - p|^r\right] \le \frac{C_{B,r}}{n^{r/2}} \tag{35}$$

*where $C_{B,r}$ is a constant depending only on $r$.*

**Proof of Lemma C.1.** The concentration inequalities (32-33) are one form of Chernoff bounds. The control (34) is Hoeffding's inequality applied to i.i.d Bernoulli random variables. Finally, for (35), see [8] or adapt the proof of Lemma C.2 below. $\qquad \square$

**Lemma C.2.** *Let $w_1, \ldots, w_n \overset{iid}{\sim} \mathrm{L}(1)$ be independent Laplace random variables with parameter $1$. Denoting the mean by $\hat{w} = \frac{1}{n} \sum_{i=1}^n w_i$, we have*

$$\mathbb{P}(\hat{w} > t) \vee \mathbb{P}(\hat{w} < -t) \le \exp\left[-\frac{n}{2}(\frac{t^2}{4} \wedge \frac{t}{2})\right]$$

$$\le \exp\left[-\frac{n}{8}t^2\right] + \exp\left[-\frac{n}{4}t\right] \quad. \tag{36}$$

*Besides, for any real $r > 0$, there exists a constant $C_{L,r} \ge 1$, depending only on $r$, such that*

$$\mathbb{E}\left(|\hat{w}|^r\right) \le \frac{C_{L,r}}{n^{r/2}} \quad. \tag{37}$$

**Proof of Lemma C.2.** A random variable $x$ is said to be sub-exponential with parameter $\lambda$, denoted $x \sim \mathrm{subE}(\lambda)$, if $\mathbb{E}\, x = 0$ and its moment generating function satisfies

$$\mathbb{E}[e^{sx}] \le e^{\lambda^2 s^2/2}, \quad \forall |s| < \frac{1}{\lambda}.$$

Let $x_1, \ldots, x_n$ be independent random variables such that $x_i \sim \mathrm{subE}(\lambda)$. Bernstein's inequality [8] entails that, for any $t > 0$, the mean $\hat{x} = \frac{1}{n} \sum_{i=1}^n x_i$ satisfies

$$\mathbb{P}(\hat{x} > t) \vee \mathbb{P}(\hat{x} < -t) \le \exp\left[-\frac{n}{2}(\frac{t^2}{\lambda^2} \wedge \frac{t}{\lambda})\right] \quad. \tag{38}$$

Then, for any real $r > 0$ we have

$$\mathbb{E}\,|\hat{x}| = \int_0^\infty \mathbb{P}(|\hat{x}|^r > t)dt = \int_0^\infty \mathbb{P}(|\hat{x}| > t^{1/r})dt \le \int_0^\infty 2e^{-\frac{nt^{2/r}}{2\lambda^2}}\, dt + \int_0^\infty 2e^{-\frac{nt^{1/r}}{2\lambda}}\, dt$$

so that, using $u = \frac{nt^{2/r}}{2\lambda^2}$ and $v = \frac{nt^{1/r}}{2\lambda}$,

$$\mathbb{E}\,|\hat{x}| \le \left(\frac{2\lambda^2}{n}\right)^{r/2} r \int_0^\infty e^{-u}u^{(r/2)-1}du \; + \; 2\left(\frac{2\lambda}{n}\right)^r r \int_0^\infty e^{-v}v^{r-1}dv$$

$$= \left(\frac{2\lambda^2}{n}\right)^{r/2} r\Gamma(r/2) \; + \; 2\left(\frac{2\lambda}{n}\right)^r r\Gamma(r)$$

$$\le 2^{r+2}\lambda^r r\left[\Gamma(r/2) + \Gamma(r)\right]\frac{1}{n^{r/2}} \quad. \tag{39}$$

Let $w \sim L(1)$ be a random variable of Laplace distribution with parameter 1. Observe that $\mathbb{P}(|w| > t) = e^{-t}$ for $t \geq 0$, and

$$\mathbb{E}[e^{sw}] \leq e^{2s^2}, \quad \text{if } |s| < \frac{1}{2}.$$

Hence, $w$ is sub-exponential with parameter 2, i.e. $w \sim \text{subE}(2)$. We can take $\lambda = 2$ in (38-39) to conclude the proof of Lemma C.2, choosing $C_{L,r} = 2^{2r+2}r\left[\Gamma(r/2) + \Gamma(r)\right]$. $\qquad\square$

## D    Proofs of lower bounds

**Proof of Proposition 2.2.**    Recall that $\hat{z}_k = \frac{1}{n}\sum_{i=1}^n z_{ik}$, where $z_{ik} = \mathbb{1}_{\{x_i = k\}} + \frac{\sigma}{\alpha} \cdot w_{ik}$, with $\mathbb{E}\,z_{ik} = p_k$ and $\text{Var}(z_{ik}) = p_k(1 - p_k) + \frac{2\sigma^2}{\alpha^2}$. Note that $\tilde{\tau} := \frac{\sigma}{\sqrt{\alpha^2 n}}$ lies in $[0,2]$, and that $\text{Var}(z_{ik}) \geq (\sqrt{n}\tilde{\tau})^2$. By the central limit theorem, $\sqrt{n}\frac{\hat{z}_k - p_k}{\sqrt{\text{Var}(z_{ik})}}$ has an asymptotic standard normal distribution, so we have $\mathbb{P}(\sqrt{n}\frac{\hat{z}_k - p_k}{\sqrt{\text{Var}(z_{ik})}} \geq 1) \geq c_1$ for some numerical constant $c_1 > 0$ and $n$ large enough. We write $\hat{z}_k = \sqrt{n}\frac{\hat{z}_k - p_k}{\sqrt{\text{Var}(z_{ik})}} \cdot \frac{\sqrt{\text{Var}(z_{ik})}}{\sqrt{n}} + p_k \geq \frac{\sqrt{\text{Var}(z_{ik})}}{\sqrt{n}}$ with probability larger than $c_1$, thus leading to

$$\mathbb{E}\left[(T_{[0,2]}(\hat{z}_k))^\gamma\right] - p_k^\gamma \geq c_1\left(T_{[0,2]}\left(\frac{\sqrt{\text{Var}(z_{ik})}}{\sqrt{n}}\right)\right)^\gamma - p_k^\gamma = c_1\tilde{\tau}^\gamma - p_k^\gamma \geq \frac{c_1\tilde{\tau}^\gamma}{2}, \qquad \text{as } n \to \infty$$

for all $p_k \leq \left(\frac{c_1}{2}\right)^{1/\gamma}\tilde{\tau}$. Denoting by $\mathcal{K}_{\leq(c_1/2)^{1/\gamma}\tilde{\tau}}$ the number of such $p_k$ satisfying the latter inequality, we get

$$\sum_{k \in \mathcal{K}_{\leq(c_1/2)^{1/\gamma}\tilde{\tau}}} \mathbb{E}\left[(T_{[0,2]}(\hat{z}_k))^\gamma\right] - p_k^\gamma \geq \frac{c_1\tilde{\tau}^\gamma|\mathcal{K}_{\leq(c_1/2)^{1/\gamma}\tilde{\tau}}|}{2}, \qquad \text{as } n \to \infty. \qquad (40)$$

Hence, the lower bound announced in Proposition 2.2 holds in particular for any $p = (p_1, \ldots, p_K) \in \mathcal{P}_K$ such that $|\mathcal{K}_{\leq(c_1/2)^{1/\gamma}\tilde{\tau}}| = K$. However, this last equality entails that $K$ satisfies the following restriction $K \gtrsim_\gamma (\tilde{\tau})^{-1} \gtrsim_\gamma \sqrt{\alpha^2 n}$ since $\sum_{k=0}^K p_k = 1$. We remove this restriction in the sequel.

Let $C > 0$ be some constant that will be set later, and that only depends on $\gamma$. If $K \leq C\left(1 \vee (\alpha^2 n)^{\frac{\gamma}{2} - \frac{1}{2}}\right)$, then the lower bound of Proposition 2.2 follows directly from Theorem 2.6. We can therefore assume that

$$K \geq C\left(1 \vee (\alpha^2 n)^{\frac{\gamma}{2} - \frac{1}{2}}\right). \qquad (41)$$

Let $p = (p_1, \ldots, p_K) \in \mathcal{P}_K$ such that $p_j \leq \left(\frac{c_1}{2}\right)^{1/\gamma}\tilde{\tau}$ for all $j \in [K-1]$, and $p_K \in [0,1]$ so that $\sum_{k=1}^K p_k = 1$. By Lemma B.5 and B.7, the bias of estimation of $p_K$ is bounded by

$$\left|\mathbb{E}\left[(T_{[0,2]}(\hat{z}_K))^\gamma\right] - p_K^\gamma\right| \leq C'\left(\frac{1}{(\alpha^2 n)^{\gamma/2}} + \mathbb{1}_{\{\gamma \geq 2\}}\frac{1}{\alpha^2 n}\right),$$

where $C'$ is a constant depending only on $\gamma$. Combining with (40), we get

$$\sum_{k=1}^K \mathbb{E}\left(T_{[0,2]}(\hat{z}_k)\right)^\gamma - p_k^\gamma \geq \frac{c_1\tilde{\tau}^\gamma(K-1)}{2} - \frac{C'}{(\alpha^2 n)^{\gamma/2}} - \mathbb{1}_{\{\gamma \geq 2\}}\frac{C'}{\alpha^2 n}$$

$$\geq \frac{c_1 K}{4(\alpha^2 n)^{\gamma/2}} - \frac{C'}{(\alpha^2 n)^{\gamma/2}} - \mathbb{1}_{\{\gamma \geq 2\}}\frac{C'}{\alpha^2 n}.$$

Hence, it suffices to choose a large enough constant $C$ in (41) to have

$$\sum_{k=1}^K \mathbb{E}\left(T_{[0,2]}(\hat{z}_k)\right)^\gamma - p_k^\gamma \geq \frac{C''K}{(\alpha^2 n)^{\gamma/2}}$$

for some constant $C''$ depending only on $\gamma$. We have proved the desired lower bound under the assumption (41). The proof of Proposition 2.2 is complete. $\qquad\square$

**Proof of Theorem 2.6.** Fix $\gamma > 0, \gamma \neq 1$. Let $\tilde{\tau} := \frac{\tilde{C}}{\sqrt{\alpha^2 n}}$ for a constant $\tilde{C} \in (0, 1)$ that will be set later, and which only depends on $\gamma$. Let us start with the case $K = 2$. Define two probability vectors $p = (p_1, p_2) = (1 - \tilde{\tau}, \tilde{\tau})$ and $q = (q_1, q_2) = (1 - \tilde{\tau}/2, \tilde{\tau}/2)$. Then for a small enough constant $\tilde{C}$, we have

$$\Delta := |F_\gamma(p) - F_\gamma(q)| = |(1 - \tilde{\tau})^\gamma - (1 - \tilde{\tau}/2)^\gamma + \tilde{\tau}^\gamma - (\tilde{\tau}/2)^\gamma|$$

$$= \left| -\frac{\gamma\tilde{\tau}}{2} + O(\tilde{\tau}^2) + \tilde{\tau}^\gamma(1 - \frac{1}{2^\gamma}) \right|$$

where we used $(1 - x)^\gamma = 1 - \gamma x + O(x^2)$ for any real $x \in (0, \tilde{C})$. If $\gamma \in (0, 1)$, we can choose $\tilde{C}$ small enough to have

$$\Delta = \tilde{\tau}^\gamma \left| -\frac{\gamma\tilde{\tau}^{1-\gamma}}{2} + O(\tilde{\tau}^{2-\gamma}) + (1 - \frac{1}{2^\gamma}) \right| \geq C\tilde{\tau}^\gamma$$

for some constant $C$ depending only on $\gamma$. Similarly, if $\gamma > 1$, we have

$$\Delta = \tilde{\tau} \left| -\frac{\gamma}{2} + O(\tilde{\tau}) + \tilde{\tau}^{\gamma-1}(1 - \frac{1}{2^\gamma}) \right| \geq C\tilde{\tau} .$$

For any $\alpha$-LDP mechanism $Q$, denote by $Qp$ and $Qq$ the measures corresponding to the channel $Q$ applied to the probability vectors $p$ and $q$. Corollary 3 of [3] ensures that the Kullback-Leibler divergence between $Qp$ and $Qq$ is bounded by

$$D_{kl}(Qp, Qq) \leq 4(e^\alpha - 1)^2 n \left( d_{TV}(p, q) \right)^2 ,$$

i.e. by $n$ times the square of the total variation distance between $p$ and $q$, up to a constant depending on $\alpha$. Then we have

$$D_{kl}(Qp, Qq) \leq 4(e^\alpha - 1)^2 n \left( \sum_{k=1}^2 |p_k - q_k| \right)^2 \leq 4(e^\alpha - 1)^2 n\tilde{\tau}^2 \leq 36\tilde{C}^2 \qquad (42)$$

where the last inequality follows from $e^x - 1 \leq 3x$ for any $x \in [0, 1]$.

For any vector $\theta = (\theta_1, \theta_2), \theta_i \geq 0$, we denote the functional at $\theta$ by $F_\gamma(\theta) = \sum_{k=1}^2 \theta_k^\gamma$. We use a standard lower bound method based on two hypotheses, see e.g. Theorem 2.1 and 2.2 in [7], to get for any estimator $\hat{F}$,

$$\sup_{\theta \in \{p,q\}} \mathbb{P}_\theta \left( |\hat{F} - F_\gamma(\theta)| \geq \frac{\Delta}{2} \right) \geq \frac{1 - \sqrt{D_{kl}(Qp, Qq)/2}}{2} .$$

Then we deduce from (42) that

$$\sup_{\theta \in \{p,q\}} \mathbb{P}_\theta \left( |\hat{F} - F_\gamma(\theta)| \geq \frac{\Delta}{2} \right) \geq \frac{1 - 3\sqrt{2}\tilde{C}}{2} \geq \frac{1}{4} ,$$

choosing $\tilde{C} \leq 1/(6\sqrt{2})$. We have proved the desired lower bound in the case $K = 2$.

We can actually prove the same lower bound for any integer $K \geq 2$, with the following slight modification in the proof written above. Choose $p_k, q_k, k \geq 3$ such that $p_k = q_k$ and $p_k \leq \tilde{C}/(4Kn)$. Then change the $p_1$ and $q_1$ above accordingly (to have probability vectors). This affects neither the order of the separation $\Delta$, nor the bound on the KL-divergence between the measures $Qp$ and $Qq$. This concludes the proof of Theorem 2.6. $\qquad \square$

**Proof of Theorem 2.7.** If $K < 4$, then the lower bounds are a direct consequence of Theorem 2.6. We assume therefore that $K \geq 4$. For the ease of exposition, we also assume that $K$ is even (the case of an odd $K$ being similar). Let $\tilde{K}$ be a positive even integer in $[K]$. Let $p = (p_1, \ldots, p_K)$ be any probability vector such that two consecutive coordinates are equal $p_{2k-1} = p_{2k}$ for $k \in [\tilde{K}/2]$, and the remaining coordinates satisfy $p_k = p_{k'}$ for all $k, k' \geq \tilde{K} + 1$. Similarly, let $\delta = (\delta_1, \ldots, \delta_K)$ be a vector of perturbations such that, two consecutive perturbations are equal $\delta_{2k-1} = \delta_{2k}, k \in [\tilde{K}/2]$,

and the others are equal to zero: $\delta_k = 0$ , $\forall\, k \geq \tilde{K} + 1$. Each perturbation is smaller than (half of) the corresponding probability: $0 \leq \delta_k \leq p_k/2$, $k \in [\tilde{K}]$. Given any $k \in [K/2]$ and any vector $q = (q_1, \ldots, q_K)$, define the operator $T_k(q) = (0, \ldots, 0, q_{2k-1}, -q_{2k}, 0, \ldots, 0)$. We are now ready to introduce the following collection of vectors $p^{(\nu)}$, $\nu \in \mathcal{V}\{-1, 1\}^{\tilde{K}/2}$:

$$
\begin{aligned}
p^{(\nu)} &= p + \sum_{k=1}^{\tilde{K}/2} \nu_k T_k(\delta) \\
&= (p_1, p_2, p_3, p_4, \ldots, p_{K-1}, p_K) + (\nu_1\delta_1, -\nu_1\delta_2, \ldots, \nu_{\tilde{K}/2}\delta_{\tilde{K}-1}, -\nu_{\tilde{K}/2}\delta_{\tilde{K}}, 0, \ldots, 0) \\
&= (p_2, p_2, p_4, p_4, \ldots, p_{\tilde{K}}, p_{\tilde{K}}, p_K, \ldots, p_K) + (\nu_1\delta_2, -\nu_1\delta_2, \ldots, \nu_{\tilde{K}/2}\delta_{\tilde{K}}, -\nu_{\tilde{K}/2}\delta_{\tilde{K}}, 0, \ldots, 0) \ .
\end{aligned}
$$

Observe that each $p^{(\nu)}$, $\nu \in \mathcal{V}\{-1, 1\}^{\tilde{K}/2}$, is a vector of probability. We bound from below the difference between $F_\gamma(p^{(\nu)})$ and $F_\gamma(p)$ in the next lemma, whose proof is postponed at the end of the section.

**Lemma D.1.** *For any $\gamma \in (0, 2)$, $\gamma \neq 1$, and any $\nu \in \mathcal{V}\{-1, 1\}^{\tilde{K}/2}$, we have*

$$
|F_\gamma(p^{(\nu)}) - F_\gamma(p)| \geq C \sum_{k=1}^{\tilde{K}/2} p_{2k}^{\gamma-2}\delta_{2k}^2 =: R
$$

*for a constant $C > 0$ depending only on $\gamma$.*

We will show that it is hard to know if the data come from $p$ or a uniform mixture of the $p^{(\nu)}$, $\nu \in \mathcal{V}$. We do so by using Theorem A.1 of [6], with the notations of [6]. For any fixed $\alpha$-LDP interactive mechanism $Q$, we write $Q^n := (Qp)^n \in \mathrm{conv}\left(Q\mathcal{P}_{\leq F_\gamma(p)}^{(n)}\right)$ and $\overline{Q}^n := 2^{-\tilde{K}/2}\sum_{\nu\in\mathcal{V}}(Qp^{(\nu)})^n \in \mathrm{conv}\left(Q\mathcal{P}_{\geq F_\gamma(p)+R}^{(n)}\right)$. With the notations of [6] and standard relations between probability metrics, we have that the upper affinity satisfies

$$
\eta_A^{(n)}(Q, R) \geq \pi(Q^n, \overline{Q}^n) = 1 - d_{TV}(Q^n, \overline{Q}^n) \geq 1 - \sqrt{D_{kl}(Q^n, \overline{Q}^n)/2} \ . \tag{43}
$$

We can bound the KL-divergence $D_{kl}(Q^n, \overline{Q}^n)$ as in the proof of Theorem 4.2 in [2], and have

$$
D_{kl}(Q^n, \overline{Q}^n) \leq \frac{n(e^{2\alpha} - e^{-2\alpha})^2}{4}\|\delta\|_2^2 \ .
$$

Hence, it suffices to choose a $\delta$ satisfying the condition

$$
\|\delta\|_2^2 \leq \frac{2}{n(e^{2\alpha} - e^{-2\alpha})^2} \ , \tag{44}
$$

to have $\eta_A^{(n)}(Q, R) \geq \frac{1}{2}$. Denoting $\Delta_A^{(n)}(Q, \eta) := \sup\{\Delta \geq 0 \,:\, \eta_A^{(n)}(Q, \Delta) > \eta\}$ as in [6], we will get for any $\eta \in (0, 1/2)$,

$$
\Delta_A^{(n)}(Q, \eta) \geq R
$$

where $R$ is defined in Lemma D.1 above. It will then follow from Theorem A.1 of [6] that

$$
\inf_Q \inf_{\hat{F}} \sup_{p\in\mathcal{P}} \mathbb{E}\left[(\hat{F} - F_\gamma(p))^2\right] \geq \left(\frac{R}{2}\right)^2 \frac{\eta}{2} \ ,
$$

for any $\eta \in (0, 1/2)$. Taking $\eta = 1/4$ we will have

$$
\inf_Q \inf_{\hat{F}} \sup_{p\in\mathcal{P}} \mathbb{E}\left[(\hat{F} - F_\gamma(p))^2\right] \geq \frac{C^2}{32}\left(\sum_{k=1}^{\tilde{K}/2} p_{2k}^{\gamma-2}\delta_{2k}^2\right)^2 \ .
$$

To choose a $\delta$ fulfilling (44), we consider two cases according to the values of $K$.

$1°$. *In the case where $K < n(e^{2\alpha} - e^{-2\alpha})^2$, we choose $\tilde{K} = K$, and take $\delta_k = (4\sqrt{Kn}(e^{2\alpha} - e^{-2\alpha}))^{-1}$, $k \in [K]$. We take $p_k = 2\delta_k$, $k \in [K-2]$, and the remaining $p_{K-1}, p_K \geq 2\delta_k$ so that $p$ is a vector of probability (i.e. $\sum_k p_k = 1$). This gives*

$$\inf_Q \inf_{\hat{F}} \sup_{p \in \mathcal{P}} \mathbb{E}\left[(\hat{F} - F_\gamma(p))^2\right] \geq \frac{C^2}{32}\left(\frac{2^{\gamma-2}[(K/2)-1]}{(4\sqrt{Kn}(e^{2\alpha}-e^{-2\alpha}))^\gamma}\right)^2 \geq \frac{C^2 2^{-2\gamma}}{8192}\frac{K^{2-\gamma}}{((e^{2\alpha}-e^{-2\alpha})^2 n)^\gamma} \ ,$$

where we used $(K/2) - 1 \geq K/4$ with $K \geq 4$. This corresponds to the right term of both lower bounds announced in Theorem 2.7.

$2°$. *In the case where $K \geq n(e^{2\alpha} - e^{-2\alpha})^2$, we separate our analysis in two ranges of values of $\gamma$. If $\gamma \in (0,1)$, we take $\tilde{K} = K$, and $\delta_k = (2K)^{-1}$ and $p_k = 2\delta_k$ for all $k \in [K]$. This leads to*

$$\inf_Q \inf_{\hat{F}} \sup_{p \in \mathcal{P}} \mathbb{E}\left[(\hat{F} - F_\gamma(p))^2\right] \geq \frac{C^2}{32}\left(\frac{2^{\gamma-2}(K/2)}{(2K)^\gamma}\right)^2 \geq \frac{C^2}{2048}K^{2(1-\gamma)} \ ,$$

which matches the first term of the lower bound for $\gamma \in (0,1)$ in the theorem.
If $\gamma \in (1,2)$, let $\tilde{K}$ be the smallest even integer satisfying $\tilde{K} \geq n(e^{2\alpha} - e^{-2\alpha})^2$ and $\tilde{K} \geq 4$. We set $\delta_k = (8\sqrt{\tilde{K}n}(e^{2\alpha} - e^{-2\alpha}))^{-1}$ for $k \in [\tilde{K}]$. We choose $p_k = 2\delta_k$ for $k \in [\tilde{K} - 2]$, and $p_k \geq 2\delta_k$ for $k \geq \tilde{K} - 1$ such that $p$ is a vector of probability. Then

$$\inf_Q \inf_{\hat{F}} \sup_{p \in \mathcal{P}} \mathbb{E}\left[(\hat{F} - F_\gamma(p))^2\right] \geq \frac{C^2}{32}\left(\frac{2^{\gamma-2}\left[(\tilde{K}/2)-1\right]}{(8\sqrt{\tilde{K}n}(e^{2\alpha}-e^{-2\alpha}))^\gamma}\right)^2$$

$$\geq \frac{C^2 4^{-2\gamma}}{8192}\frac{\tilde{K}^{2-\gamma}}{((e^{2\alpha}-e^{-2\alpha})^2 n)^\gamma}$$

$$\geq \frac{C^2 4^{-2\gamma}}{8192}((e^{2\alpha}-e^{-2\alpha})^2 n)^{2(1-\gamma)} \ ,$$

which corresponds to the first term of the lower bound for $\gamma \in (1,2)$ in the theorem.

The proof of Theorem 2.7 is complete. $\qquad\square$

**Proof of Lemma D.1.** We have

$$F_\gamma(p^{(\nu)}) - F_\gamma(p) = \sum_{k=1}^{\tilde{K}/2}\left[(p_{2k} + \nu_k\delta_{2k})^\gamma + (p_{2k} - \nu_k\delta_{2k})^\gamma - 2p_{2k}^\gamma\right] \tag{45}$$

Denoting $f(x) = x^\gamma$ and using Taylor's formula, we have for any real $Y > 0$,

$$f(Y) = f(p_{2k}) + f'(p_{2k})(Y - p_{2k}) + f''(w_Y)\frac{(Y - p_{2k})^2}{2}$$

where $w_Y$ lies between $Y$ and $p_{2k}$. We take $Y = p_{2k} + \nu_k\delta_{2k}$ and $\tilde{Y} = p_{2k} - \nu_k\delta_{2k}$ to get

$$f(Y) + f(\tilde{Y}) - 2p_{2k}^\gamma = f''(w_Y)\frac{(Y - p_{2k})^2}{2} + f''(w_{\tilde{Y}})\frac{(Y - p_{2k})^2}{2}$$

$$= \gamma(\gamma-1)(w_Y^{\gamma-2} + w_{\tilde{Y}}^{\gamma-2})\frac{\delta_{2k}^2}{2} \tag{46}$$

Since $w_Y \vee w_{\tilde{Y}} \leq p_{2k} + \delta_{2k}$ with $0 \leq \delta_{2k} \leq p_{2k}/2$, and $\gamma \in (0,2)$, we have

$$w_Y^{\gamma-2} \wedge w_{\tilde{Y}}^{\gamma-2} \geq (p_{2k} + \delta_{2k})^{\gamma-2} \geq (2p_{2k})^{\gamma-2} \ .$$

Hence, for $\gamma \in (1,2)$,

$$f(Y) + f(\tilde{Y}) - 2p_{2k}^\gamma \geq \gamma(\gamma-1)(2p_{2k})^{\gamma-2}\delta_{2k}^2$$

which leads to the desired lower bound of (45). For $\gamma \in (0,1)$, we deduce from (46) that all terms of the sum (45) are non-positive and satisfy

$$f(Y) + f(\tilde{Y}) - 2p_{2k}^\gamma \leq \gamma(\gamma-1)(2p_{2k})^{\gamma-2}\delta_{2k}^2 \ .$$

So, the absolute value of the sum (45) can be lower bounded as announced in the lemma. $\qquad\square$