# OpenReview forum: "Locally differentially private estimation of functionals of discrete distributions"
_NeurIPS.cc/2021/Conference — NeurIPS 2021 Poster_

### Official Review · Reviewer_H2n9 · 2021-07-16

**Rating:** 5
**Confidence:** 5

**Summary:**

This paper studies the problem of estimating non-linear functionals of discrete distributions
in the  local differential privacy model. The authors first study the non-interactive case and provide both lower and upper bound of the minimax private rate. Then they showed that in the interactive case there is an algorithm that is much more better than the optimal minimax private rate in the non-interactive one.

**Limitations And Societal Impact:**

I tend to weakly reject the paper. My main reason is the topic itself is too narrow to ML or DP audience, and I think it will better to submit the paper to information theory or statistics area.

1) As I see from the reference of this paper, in the previous ICML or NeurIPS conferences. There is no paper on estimating non-linear functionals of discrete distributions and most of the related papers appeared at information theory or statistics area. So I do not think NeurIPS is a good place to this paper.

2) As the authors mentioned, this is the first paper studying the problem in LDP model. My question is why the authors study LDP model first instead of the central model? Is that local model is easier?

3) There is no experimental study.

4) The motivation of this paper is unclear. The authors should claim this problem indeed has potential privacy issue (for example, a previous work indeed studied the problem with some sensitive data)

**Main Review:**

Originality: This paper provides the first study on the problem of estimating non-linear functionals of discrete distributions
in the  local differential privacy model.

Quality: The paper is a theoretical paper, some of the proofs are tricky which might be used to other problems. Moreover, they provide both lower and upper bounds. And show the advantage if we allow interactions in the LDP model.

Although the proof is hard to following, the idea of the paper is clear and readable.

**Time Spent Reviewing:**

6h

---

> ### Author Response · Authors · 2021-08-09
> **Motivation, numerical experiments, and updated manuscript**
>
> We thank the referee for his/her careful reading of our manuscript and numerous constructive remarks and questions.
>
> $1^{\circ}$, $2^{\circ}$ and $4^{\circ}$. Motivation:
>
> Indeed, our previous manuscript did not expand on the applications and practical motivation for studying this problem. We will include such a discussion together with the numerous references that study this problem in the non-private case and in the global DP setup. We will insist on major difficulties and changes when local DP is considered.  A significant number of these references are from Neurips.
>
> Typical motivation for such functionals is to estimate the global amount of information criteria of the probability distribution. The entropy of a probability distribution with finite support and probabilities bounded away from 0 can be approximated via a Taylor expansion to a linear combination of $F_\gamma$ for integer values of $\gamma$. Rényi entropies also involve these functionals.
>
> It is known that symmetric functions of $p_1,\ldots,p_K$, i.e. functions of at most $K$ variables that are permutation invariant, can be written as a polynomial of functionals $F_\gamma$ for positive integer values of $\gamma$.
>
> Another important application of such functionals is hypothesis testing. When the distance between 2 probability distributions $p$ and $p_0$ is evaluated by a discrepancy or a distance, functionals $F_\gamma$ naturally appear in their expression. For example, calculating the Hellinger distance to the uniform distribution $\sum_{j=1}^K (\sqrt{p_j} - \frac 1{\sqrt{K}})^2$, involves estimating $F_{1/2}$ for $\gamma = 1/2$. Thus, an uniformity test procedure will be based on the estimator of this discrepancy, and similarly for identity or closeness tests. However, testing rates may differ from the esimation rates as we know that several distances can be agregated to produce an optimal testing procedure (e.g. Valiant and Valianet 2017).
>
>
>
> We believe that it is interesting to assess the performance of simple estimators that are widely used and very easy to implement in the non private case, in order to understand the differences between LDP and non-private settings. We have thus started the study of the power sum functional with a plug-in estimator. Although such estimator  is known to be nearly minimax optimal in the non private case (up to a poly-logarithmic factor- see [14]),  we show that it performs poorly in the LDP setting (proving a tight characterization of its maximal quadratic risk, with both an upper bound and a matching lower bound). This highlights an important difference between LDP and non private setting: practical methods performing well in the non private case should not be systematically transferred to the LDP setting. As a complement, we suggest a correction of this plug-in estimator using a standard threshold method, which significantly improves the rates of convergence. We emphasize that the improvement is significant: it consists in removing the dependence of the risk in the support size $K$, when $K$ is large. This is notably different from gaining logarithmic factors, as it is often the issue in the literature on functional estimation.
>
>
>
>
> $3^{\circ}$. We agree that numerical experiments will add value to the paper.  We expect our simple methods to work quite fast and well when compared to the non-private case. Due to the limited space, we do not think that a complete numerical study could be added (several values of $\gamma$, several estimators, etc.), only maybe a brief one.\\
>
>
>
>
> Updated manuscript: We have already prepared a new version of the paper that is hopefully improved by both clarifying the global presentation of our results and by slightly improving some results:
>
> (i) We have added a precise high level presentation of our results in Section~2. One theorem states the best convergence rates obtained by combining our different estimators according to the values of the problem's parameters  (support size $K$ of the underlying distribution $(p_1,\ldots,p_K)$, and power $\gamma$ of the functional $F_{\gamma}$). We discuss the phase transitions between these rates, then we outline  the rest of the paper and the relations between the sections.
>
> (ii) All the loose statements about feasible extensions and conjectures have been either deleted or clarified and re-located in the last section (discussion).
>
> (iii) We rewrote  Section 3.3 about the thresholded plug-in estimator. After having revisited the proof of Theorem 3.4, we now give a sharper error bound, which is now a minimum between two error bounds: the error bound already given (in this theorem) in the first version of the paper, and the error bound of the plug-in estimator (up to some logarithmic factor). This new bound is therefore equal to the former bound when the support size $K$ is large, and to the latter bound when $K$ is small ($K\lesssim \sqrt{\alpha^2 n}$).
> This shows that the thresholded plug-in estimator improves the performance of the plug-in estimator for large $K$, while conserving its performance for small $K$ (up to some logarithmic factor).
>
>
> (iv) We have improved the lower bound in Theorem 5.2, separating the two cases  $\gamma \in (0,1)$ and  $\gamma \in (1,2)$. For $\gamma \in (0,1)$, we have kept the same lower bound as in the first version of the paper. For $\gamma \in (1,2)$, we have replaced the first term  $K^{2(1-\gamma)}$ of the lower bound by $(\alpha^2 n)^{2(1-\gamma)}$. Recall that the lower bound is a minimum between two terms, where the first term corresponds to the regime $K \geq \alpha^2 n$. Since $K^{2(1-\gamma)}\leq (\alpha^2 n)^{2(1-\gamma)}$ in this regime, our replacement of the first term  improves the lower bound. This result follows from  a slight  adjustment in the proof of the theorem.

---

### Official Review · Reviewer_ztVm · 2021-07-18

**Rating:** 5
**Confidence:** 4

**Summary:**

The paper considers the problem of estimating the power sum of discrete distributions p under local differential privacy (LDP) constraints, which is defined as $F_\gamma(p) = \sum_x p_x^{\gamma}$. The paper analyzes three types of algorithms.

The first is the plug-in estimator which first estimates the distribution p by $\hat p$ using Laplace mechanism and then the final estimate is simply $F_{\gamma}(\hat p)$. The paper characterizes the minimax mean squared error (MSE) for the plug-in estimator for $\gamma \in (0,1)$.

The second is a thresholded plug-in estimator which sets a threshold and the final estimate is only the sum of functionals over heavy elements. The paper shows that the estimator can achieve an MSE which depends only logarithmically on the alphabet size K.

Both above algorithms are non-interactive in the sense that each participant cannot see the message of other participants. The paper then presents an interactive algorithm that improves over the thresholded estimator by a logarithmic factor.

The paper also provides lower bounds which are off the upper bounds by a polynomial factor.

**Limitations And Societal Impact:**

Yes.

**Main Review:**

The paper considers an interesting problem and makes some nontrivial progress. However, the technical contribution still appears limited.

1. The paper focuses on analyzing plug-in estimators and variants of them. In particular, the thresholded estimator can still be seen as a plug-in estimator with the estimated distribution being the thresholded distribution. The lower bound and upper bound provided in the paper don't match hence the paper does not provide a clear scope of how well plug-in estimators perform in general. For more general interactive estimators, the provided upper and lower bounds don't match as well. Hence the contribution of these results to the understanding of the problem is limited. The paper will be stronger if the paper can provide a clearer scope of the problem by closing these gaps.

2. The writing of the paper can be improved. The paper states many results in the main paper and it is less clear to the readers what are the main theorems to look at. It would be better if the authors can emphasize some important results for better clarity.

Overall, I think the contribution of the paper is limited hence I would tend towards a rejection.

**Time Spent Reviewing:**

4

---

> ### Author Response · Authors · 2021-08-09
> **Lower/upper bounds gap and the writing of the paper**
>
> We thank the referee for his/her careful reading of our manuscript and numerous constructive remarks and questions.
>
>
> $1^{\circ}$. The loose statements have been deleted  or clarified and delayed to the last section (discussion). Please see the paragraph "Updated manuscript" below.
>
> Upper/lower bounds gap:
>
> Finding the minimax rates of estimation is of course a crucial objective; in that direction it would be natural to study the estimators based on the best polynomial approximation that are already known to be minimax optimal in the non-private case. Before tackling this task, we believe that it is interesting to assess the performance of simple estimators that are widely used and very easy to implement in the non private case, in order to understand the differences between LDP and non-private settings. We have thus started the study of the power sum functional with a plug-in estimator. Although such estimator  is known to be nearly minimax optimal in the non private case (up to a poly-logarithmic factor- see [14]),  we show that it performs poorly in the LDP setting (proving a tight characterization of its maximal quadratic risk, with both an upper bound and a matching lower bound). This highlights an important difference between LDP and non private setting: practical methods performing well in the non private case should not be systematically transferred to the LDP setting. As a complement, we suggest a correction of this plug-in estimator using a standard threshold method, which significantly improves the rates of convergence. We emphasize that the improvement is significant: it consists in removing the dependence of the risk in the support size $K$, when $K$ is large. This is notably different from gaining logarithmic factors, as it is often the issue in the literature on functional estimation.
>
> We conjecture that our upper bounds are tight, up to some logarithmic factors. In future work, this logarithmic factor should be removed, using the best polynomial approximation. On the other hand, we provide universal lower bounds with respect to all privacy mechanisms and all estimators.  This gives a point of comparison for the performances of the plug-in type estimators.  Leaving aside the logarithmic factors, we conjecture that these lower bounds are not optimal, up to a factor $K^{\gamma}$. To gain this factor $K^{\gamma}$, we may unfortunately miss a key tool in locally differential privacy, such as the two fuzzy hypothesis theorem  in the non-private case. These non-sharp lower bounds illustrate well the fact that one of the most important challenge in locally differential privacy is to provide a turnkey toolbox that help prove universal lower bounds.
>
>
> $2^{\circ}$. Updated manuscript: We have already prepared a new version of the paper that is hopefully improved by both clarifying the global presentation of our results and by slightly improving some results:
>
> (i) We have added a precise high level presentation of our results in Section 2. One theorem states the best convergence rates obtained by combining our different estimators according to the values of the problem's parameters  (support size $K$ of the underlying distribution $(p_1,\ldots,p_K)$, and power $\gamma$ of the functional $F_{\gamma}$). We discuss the phase transitions between these rates, then we outline  the rest of the paper and the relations between the sections.
>
> (ii) All the loose statements about feasible extensions and conjectures have been either deleted or clarified and re-located in the last section (discussion).
>
> (iii) We rewrote  Section 3.3 about the thresholded plug-in estimator. After having revisited the proof of Theorem 3.4, we now give a sharper error bound, which is now a minimum between two error bounds: the error bound already given (in this theorem) in the first version of the paper, and the error bound of the plug-in estimator (up to some logarithmic factor). This new bound is therefore equal to the former bound when the support size $K$ is large, and to the latter bound when $K$ is small ($K\lesssim \sqrt{\alpha^2 n}$).
> This shows that the thresholded plug-in estimator improves the performance of the plug-in estimator for large $K$, while conserving its performance for small $K$  (up to some logarithmic factor).
>
>
> (iv)  We improved the lower bound in Theorem 5.2, separating the two cases  $\gamma \in (0,1)$ and  $\gamma \in (1,2)$. For $\gamma \in (0,1)$, we kept the same lower bound as in the first version of the paper. For $\gamma \in (1,2)$, we replaced the first term  $K^{2(1-\gamma)}$ of the lower bound by $(\alpha^2 n)^{2(1-\gamma)}$. Recall that the lower bound is a minimum between two terms, where the first term corresponds to the regime $K \geq \alpha^2 n$. Since $K^{2(1-\gamma)}\leq (\alpha^2 n)^{2(1-\gamma)}$ in this regime, our replacement of the first term  is an improvement of the lower bound. This result follows from  a slight  adjustment in the proof of the theorem.

---

> > ### Comment · Reviewer_ztVm · 2021-09-01
> > **After rebuttal**
> >
> > Thanks for the response. My score on the paper remains unchanged.

---

### Official Review · Reviewer_V6UM · 2021-07-28

**Rating:** 5
**Confidence:** 5

**Summary:**

The paper studies estimating power sums of discrete distributions under local DP constraints. In particular, the results leverage standard Laplacian DP procedures and cover both the interactive and non-interactive cases. A summary of the main results appears on page 9, Table 1, for power sums with different order parameters.

**Limitations And Societal Impact:**

The authors explained the limitation of their work in the discussion section and clearly indicated (Checklist) that the work has no potential negative societal impact.

**Main Review:**

Pros:

- The paper reads well and clearly explains its relation to some of the existing works. The derivations in the appendix also seem nontrivial, presented with concise notations.



- As line 83 claims, the paper "for the first time the estimation of nonlinear and nonquadratic power-sum functionals..." In other words, I like the submission for the problem's novelty, though the approach is similar to [4] and some other existing works.



- The upper bounds mimic those non-private ones, such as [14], and demonstrate the effects of privacy constraints on the estimation rates.



- In the discussion section, the authors pointed out the limitations of their results and proposed several exciting directions for future research. I appreciate the authors for their honesty and hopefulness.





Cons:


- The paper misses numerous essential references in the "Related Literature" section:

  - Estimating functionals of discrete distributions under DP constraints have been studied in the following two papers. The first paper considers specific cases such as entropy, and the second paper further addresses all Lipschitz additive functionals.

    1. Acharya, J., Kamath, G., Sun, Z., & Zhang, H. (2018, July). "INSPECTRE: Privately Estimating the Unseen." In International Conference on Machine Learning (pp. 30-39). PMLR.

    2. Hao, Y., & Orlitsky, A. (2019). "Unified Sample-Optimal Property Estimation in Near-Linear Time." Advances in Neural Information Processing Systems, 32, 11106-11116.

  - In addition to [1], the best results for Renyi entropy estimation appear in the following two papers. The first paper considers constant accuracy and error, and the second paper studies all levels of accuracy and error.

    1. Obremski, M., & Skorski, M. (2017). "Renyi Entropy Estimation Revisited." Approximation, Randomization, and Combinatorial Optimization. Algorithms and Techniques.

    2. Hao, Y., & Orlitsky, A. (2019). "The Broad Optimality of Profile Maximum Likelihood." Advances in Neural Information Processing Systems, 32, 10991-11003.

  - The content from line 54 to line 60 does not correspond to any of the obtained results but the ones in [13, 14]. I believe it is necessary to include relevant citations in this paragraph.



- The gap between the upper and lower bounds is relatively large since the improvements are often just logarithmic for the line of works concerning distribution functionals. Also, the paper provides no concrete results about entropy estimation under DP constraints but emphasizes a few times that the technique extends to such cases. I don't find these statements very convincing in a theoretical paper. In particular, the title may oversell the results as they are just about power sums.



- The techniques used in the paper are variants of those in [2, 4, 5, 17] and are of medium novelty. I think the authors may compensate for this aspect by adding some numerical experiments on the proposed schemes. For example, showing the differences in error rates among the best estimators for non-private, non-interactive LDP, and interactive LDP cases would be excellent.




Minors:

- Lines 19 to 23, it might be better to add some references; line 68, typo in "multinomial"; line 99, typo in "mimic"; in several places, maybe write "logarithmic factor" instead of "$\log$ factor"?

**Time Spent Reviewing:**

5

---

> ### Author Response · Authors · 2021-08-09
> **Motivation and discusion on the upper/lower bounds gap**
>
> We thank the referee for his/her careful reading of our manuscript and numerous constructive remarks and questions.
>
> $1^{\circ}$. Motivation and comparison to the previous literature:
>
> We thank the reviewer V6UM for his/her suggestions. It is indeed very appropriate to cite the numerous results that have been obtained in central (or global) DP and non-private settings, to underline the major differences and additional difficulties related to the local DP setup. We will add such references.
>
>
> We agree that our former manuscript skipped this part completely.
> Typical motivation for such functionals is to estimate the global amount of information criteria of the probability distribution. It is known in the statistical literature that smooth nonlinear functionals can be reduced via Taylor expansion to estimating several functionals of the type $F_\gamma$ for positive integer values of $\gamma$ (see e.g. Giné and Nickl 2021). For example, the entropy of a probability distribution with finite support and probabilities bounded away from 0 can be approximated via a Taylor expansion to a linear combination of $F_\gamma$ for integer values of $\gamma$. Rényi entropies also involve these functionals. In order to get optimal general results for entropies on large (or infinite) dictionaries the best polynomial approximation method combined with the Laplace method could be applied (see discussion on optimality later on).
>
> It is known that symmetric functions of $p_1,\ldots,p_K$, i.e. functions of at most $K$ variables that are permutation invariant, can be written as a polynomial of functionals $F_\gamma$ for positive integer values of $\gamma$.
>
> Another important application of such functionals is hypothesis testing. When the distance between 2 probability distributions $p$ and $p_0$ is evaluated by a discrepancy or a distance, functionals $F_\gamma$ naturally appear in their expression. For example, calculating the Hellinger distance to the uniform distribution $\sum_{j=1}^K (\sqrt{p_j} - \frac 1{\sqrt{K}})^2$, involves estimating $F_{1/2}$ for $\gamma = 1/2$. Thus, an uniformity test procedure will be based on the estimator of this discrepancy, and similarly for identity or closeness tests. However, testing rates may differ from the estimation rates as we know that several distances can be aggregated to produce an optimal testing procedure (e.g. Valiant and Valiant 2017).
>
>
> $2^{\circ}$. The loose statements have been deleted  or clarified and delayed to the last section (discussion). Please see the paragraph "Updated manuscript".
>
> Upper/lower bounds gap:
>
> Finding the minimax rates of estimation is of course a crucial objective; in that direction it would be natural to study the estimators based on the best polynomial approximation that are already known to be minimax optimal in the non-private case (see the second next paragraph for this method). Before tackling this task, we believe that it is interesting to assess the performance of simple estimators that are widely used and very easy to implement in the non private case, in order to understand the differences between LDP and non-private settings. We have thus started the study of the power sum functional with a plug-in estimator. Although such estimator  is known to be nearly minimax optimal in the non private case (up to a poly-logarithmic factor- see [14]),  we show that it performs poorly in the LDP setting (proving a tight characterization of its maximal quadratic risk, with both an upper bound and a matching lower bound). This highlights an important difference between LDP and non private setting: practical methods performing well in the non private case should not be systematically transferred to the LDP setting. As a complement, we suggest a correction of this plug-in estimator using a standard threshold method, which significantly improves the rates of convergence. We emphasize that the improvement is significant: it consists in removing the dependence of the risk in the support size $K$, when $K$ is large. This is notably different from gaining logarithmic factors, as it is often the issue in the literature on functional estimation (see the second next paragraph for sophisticated estimators gaining a logarithmic factor).
>
> We conjecture that our upper bounds are tight, up to some logarithmic factors. In future work, this logarithmic factor should be removed, using the best polynomial approximation - see answer $3^{\circ}$ below. On the other hand, we provide universal lower bounds with respect to all privacy mechanisms and all estimators.  This gives a point of comparison for the performances of the plug-in type estimators.  Leaving aside the logarithmic factors, we conjecture that these lower bounds are not optimal, up to a factor $K^{\gamma}$. To gain this factor $K^{\gamma}$, we may unfortunately miss a key tool in locally differential privacy, such as the two fuzzy hypothesis theorem  in the non-private case. These non-sharp lower bounds illustrate well the fact that one of the most important challenge in locally differential privacy is to provide a turnkey toolbox that help prove universal lower bounds.
>
>
> $3^{\circ}$. Best polynomial approximation:
>
> We are aware that the minimax rate in the non-private case can be achieved using the best polynomial approximation of the non-smooth part of the functional. For the power sum, we would expect rates faster by a logarithmic factor than ours. The method amounts to identifying the small $p_k$ in the underlying distribution $(p_1,\ldots,p_K)$ using empirical frequencies $\hat{p}_k$,  and estimate the associated functional parts $p_k^{\gamma}$ using $P_d(\hat{p}_k)$ where $P_d$ is the best polynomial approximation of degree at most $d$ of the function $x \mapsto x^{\gamma}$ on some well chosen interval $[0,\nu]$. The polynomial degree must be chosen sufficiently large to minimize the bias, but also sufficiently small to keep the variance of $P_d(\hat{p}_k)$ small. In addition, $\nu$ should be small  enough to have a small bias, but also large enough to be able to identify the $p_k\in [0, \nu]$ (non-smooth part) from the $p_k > \nu$ (smooth part), using empirical frequencies $\hat{p}_k$. The smooth part of the functional corresponds to the $p_k^{\gamma}$ with large $p_k > \nu$. If $\hat{p}_k$ is large enough to ensure that $p_k$ belongs to the smooth part, then $p_k^{\gamma}$ is estimated with a plug-in estimator $\hat{p}_k^{\gamma}$, or often with a bias-corrected plug-in estimator (see [11] to [14],  for instance).
>
> This method will be the object of further investigation mainly because the gain of a logarithmic factor did not seem (to us) the most pressing question in the LDP estimation of $F_{\gamma}$, for now. Indeed, there are still missing pieces in the comprehension of the problem, mainly proving lower bounds that do not suffer from a factor $K^{\gamma}$.
>
>
>
>
> Numerical experiments:
>
> We agree that numerical experiments will add value to the paper.  We expect our simple methods to work quite fast and well when compared to the non-private case. Due to the limited space, we do not think that a complete numerical study could be added (several values of $\gamma$, several estimators, etc.), only maybe a brief one.
>
> $4^{\circ}$. Thank you for your careful reading, we will fix these minor errors.\\
>
>
>  Updated manuscript: We have already prepared a new version of the paper that is hopefully improved by both clarifying the global presentation of our results and by slightly improving some results:
>
> (i) We have added a precise high level presentation of our results in Section 2. One theorem states the best convergence rates obtained by combining our different estimators according to the values of the problem's parameters  (support size $K$ of the underlying distribution $(p_1,\ldots,p_K)$, and power $\gamma$ of the functional $F_{\gamma}$). We discuss the phase transitions between these rates, then we outline  the rest of the paper and the relations between the sections.
>
> (ii) All the loose statements about feasible extensions and conjectures have been either deleted or clarified and re-located in the last section (discussion).
>
> (iii) We rewrote  Section 3.3 about the thresholded plug-in estimator. After having revisited the proof of Theorem 3.4, we now give a sharper error bound, which is now a minimum between two error bounds: the error bound already given (in this theorem) in the first version of the paper, and the error bound of the plug-in estimator (up to some logarithmic factor). This new bound is therefore equal to the former bound when the support size $K$ is large, and to the latter bound when $K$ is small ($K\lesssim \sqrt{\alpha^2 n}$).
> This shows that the thresholded plug-in estimator improves the performance of the plug-in estimator for large $K$, while conserving its performance for small $K$  (up to some logarithmic factor).
>
>
> (iv)  We improved the lower bound in Theorem 5.2, separating the two cases  $\gamma \in (0,1)$ and  $\gamma \in (1,2)$. For $\gamma \in (0,1)$, we kept the same lower bound as in the first version of the paper. For $\gamma \in (1,2)$, we replaced the first term  $K^{2(1-\gamma)}$ of the lower bound by $(\alpha^2 n)^{2(1-\gamma)}$. Recall that the lower bound is a minimum between two terms, where the first term corresponds to the regime $K \geq \alpha^2 n$. Since $K^{2(1-\gamma)}\leq (\alpha^2 n)^{2(1-\gamma)}$ in this regime, our replacement of the first term  is an improvement of the lower bound. This result follows from  a slight  adjustment in the proof of the theorem.

---

### Official Review · Reviewer_zEEc · 2021-07-28

**Rating:** 7
**Confidence:** 4

**Summary:**

The paper studies the problem of estimating power sum functionals  F_{\gamma}:= \sum_i p_i^\gamma for \gamma>0 using locally differentially private (LDP) samples and derives the bounds on the mean square error (MSE).
The results for \gamma =2 follows from [2] and [4], and this paper derives the results for general \gamma.

To obtain the upper bound, authors privatize the samples using the standard Laplace mechanism, where each sample can be thought of as a collection of indicator random variables.
The average of these privatized indicator random variables forms an unbiased estimator of the distribution vector, which is then used in the plug-in estimator of F_{\gamma}.
For large alphabet size, they obtain a better bound on MSE by detecting and zeroing small probabilities.
Further, the authors consider a procedure similar to the interactive procedure used in [4].
For this interactive procedure, the upper bound obtained on the MSE error is smaller by log factors than the non-interactive thresholded estimator.

The authors also provide the lower bounds that match the upper bound for \gamma \ge 2.



**Limitations And Societal Impact:**

I do not see any potential negative impact of the work on society.

**Main Review:**

The problem of estimating the power sum of a discrete distribution has recently received a lot of attention
and is now fairly well understood for non-private samples.
For LDP samples the paper extends the previous results beyond the special case of \gamma =2.
The upper bounds are derived using the now-standard Laplace mechanism and plug-in-based estimators.
The lower bounds are for all possible privacy mechanisms and over all possible estimators.
The lower and upper bound match for \gamma\ge 2, but they may differ by polynomial factors for \gamma<1.

The paper is generally well written and appropriate references are provided.
All the proofs appear in the appendix.
For the lower bounds in section 5, and no intuition is provided in the main paper.
It would beneficial for the readers to include at least a brief outline of these proofs in the main paper.


Other comments:
Please fix the reference [2].
It is unclear to me if statements in lines 302 and 173 are facts or mere conjectures, in either case, these statements should be made more precise.

**Time Spent Reviewing:**

6

---

> ### Author Response · Authors · 2021-08-09
> **Modifications (done or to be done)**
>
> We thank the referee for his/her careful reading of our manuscript and numerous constructive remarks and questions.
>
> 1. Thank you for this important remark. We will give the rationale for our results, and give a sketch of proof in the main paper.
>
> 2. We have corrected our citations using the bibtex provided by NeurIPS 2020. Also, the statement 173 has been removed in the new version of the paper, it is now presented at the end of the paper (section 6) as a future direction of research. (see section "anouvelle version" below). We will clarify the status of the statement in line 302, as a conjecture.\\
>
>
>  Updated manuscript: We have already prepared a new version of the paper that is hopefully improved by both clarifying the global presentation of our results and by slightly improving some results:
>
> (i) We have added a precise high level presentation of our results in Section 2. One theorem states the best convergence rates obtained by combining our different estimators according to the values of the problem's parameters  (support size $K$ of the underlying distribution $(p_1,\ldots,p_K)$, and power $\gamma$ of the functional $F_{\gamma}$). We discuss the phase transitions between these rates, then we outline  the rest of the paper and the relations between the sections.
>
> (ii) All the loose statements about feasible extensions and conjectures have been either deleted or clarified and re-located in the last section (discussion).
>
> (iii) We rewrote  Section 3.3 about the thresholded plug-in estimator. After having revisited the proof of Theorem 3.4, we now give a sharper error bound, which is now a minimum between two error bounds: the error bound already given (in this theorem) in the first version of the paper, and the error bound of the plug-in estimator (up to some logarithmic factor). This new bound is therefore equal to the former bound when the support size $K$ is large, and to the latter bound when $K$ is small ($K\lesssim \sqrt{\alpha^2 n}$).
> This shows that the thresholded plug-in estimator improves the performance of the plug-in estimator for large $K$, while conserving its performance for small $K$  (up to some logarithmic factor).
>
>
> (iv)  We improved the lower bound in Theorem 5.2, separating the two cases  $\gamma \in (0,1)$ and  $\gamma \in (1,2)$. For $\gamma \in (0,1)$, we kept the same lower bound as in the first version of the paper. For $\gamma \in (1,2)$, we replaced the first term  $K^{2(1-\gamma)}$ of the lower bound by $(\alpha^2 n)^{2(1-\gamma)}$. Recall that the lower bound is a minimum between two terms, where the first term corresponds to the regime $K \geq \alpha^2 n$. Since $K^{2(1-\gamma)}\leq (\alpha^2 n)^{2(1-\gamma)}$ in this regime, our replacement of the first term  is an improvement of the lower bound. This result follows from  a slight  adjustment in the proof of the theorem.

---

### Official Review · Reviewer_zYFp · 2021-07-30

**Rating:** 6
**Confidence:** 4

**Summary:**

The paper studies the problem of estimating non-linear functionals of discrete distribution under local differential privacy. Given i.i.d. data from an unknown discrete distribution, the paper wants to estimate the power sum of functional F_\gamma = \sum p_k^\gamma. The author has made the following contributions:

1. Based on the Laplace mechanism, the first algorithm is proposed and studied, as an analogue of the MLE in the multinomial model.
2. Noticing that small probabilities below the fluctuation level of their unbiased estimator need not in fact be estimated, since the bias introduced by skipping those values is smaller. The paper proposes another algorithm based on the thresholding, improving the rate of the first technique when K is large.
3. In the interactive case, the paper proposes a two-step algorithm, which removes the log factor in (2).
4. A lower bound is proposed.

**Limitations And Societal Impact:**

Did not see any negative social impact.

**Main Review:**

The paper studies the problem of estimating non-linear functionals of discrete distribution under local differential privacy. This topic is interesting. However, I do not think the paper has made a lot of theoretical contributions to the community. The Laplace mechanism and thresholding method are quite common, broadly used in the property estimation problem. Besides, in the non-private case, the optimal performance is attained by the best polynomial estimator when gamma is small. However, I do not see any effort spent to see its performance in the private case. Furthermore, the rate of the paper is not tight with respect to K when gamma is small.

Minor comments:
1. There are several papers working on the central DP property estimation, e.g., [1] and [2], which also use the best polynomial estimation. Maybe worth adding them to the reference.

[1]: http://proceedings.mlr.press/v80/acharya18a.html
[2]: https://papers.nips.cc/paper/2019/file/f06ae085fe74cd78ad5e89496b197fe1-Paper.pdf

**Time Spent Reviewing:**

3 hours

---

> ### Author Response · Authors · 2021-08-09
> **Motivation and discussion on the upper/lower bounds gap**
>
> We thank the referee for his/her careful reading of our manuscript and numerous constructive remarks and questions.
>
> $1^{\circ}$. The Laplace mechanism and the plug-in and thresholded estimators are indeed quite common because they are simple to analyze and to implement. These are convenient points from an applied point of view and also intensively used in global DP.
> Our goal is to establish recipes for users with theoretical guarantees
>
> In the non-private case, estimation of power-sum functionals was also first studied for the widely spread MLE estimator (even though sub-optimal) in [14]. The estimation rates have been later tightened to optimal bounds using the best polynomial approximation of the power function around 0. On the one hand, we proceed similarly to the non-private case and start with the analogue of the MLE that becomes the plug-in estimator. We show that its behavior is even less acceptable in the private case and show how to improve it at low computational cost.
> On the other hand, for the case $\gamma = 2$, the association of plug-in and Laplace mechanism proved to be optimal among all non-interactive mechanisms, whereas an interactive procedure has been proved to improve dramatically the minimax rates in [2]. We also tried to further understand when this phenomena still hold for different functionals that can be less smooth, or not convex, for small values of $\gamma$.
>
>
>
> We find that our rates are minimax optimal for all $\gamma >2$, and for $\gamma \in (0,1)$ when the dictionary is large enough to satisfy $n \alpha^2 \lesssim K$. A polynomial gap of order $K^\gamma$ appears otherwise. See also next point.
>
>
> $2^{\circ}$. Best polynomial approximation:
>
> We are indeed aware that the minimax rate in the non-private case can be achieved using the best polynomial approximation of the non-smooth part of the functional. For the power sum, we would expect faster rates  by a logarithmic factor than our rates. The method amounts to identifying the small $p_k$ in the underlying distribution $(p_1,\ldots,p_K)$ using empirical frequencies $\hat{p}_k$,  and estimate the associated functional parts $p_k^{\gamma}$ using $P_d(\hat{p}_k)$ where $P_d$ is the best polynomial approximation of degree at most $d$ of the function $x \mapsto x^{\gamma}$ on some well chosen interval $[0,\nu]$. The polynomial degree must be chosen sufficiently large to minimize the bias, but also sufficiently small to keep the variance of $P_d(\hat{p}_k)$ small. In addition, $\nu$ should be small  enough to have a small bias, but also large enough to be able to identify the $p_k\in [0, \nu]$ (non-smooth part) from the $p_k > \nu$ (smooth part), using empirical frequencies $\hat{p}_k$. The smooth part of the functional corresponds to the $p_k^{\gamma}$ with large $p_k > \nu$. If $\hat{p}_k$ is large enough to ensure that $p_k$ belongs to the smooth part, then $p_k^{\gamma}$ is estimated with a plug-in estimator $\hat{p}_k^{\gamma}$, or often with a bias-corrected plug-in estimator (see [11] to [14],  for instance).
>
> This method will be the object of further investigation mainly because the gain of a logarithmic factor did not seem (to us) the most pressing question in the LDP estimation of $F_{\gamma}$, for now. Indeed, there are still missing pieces in the comprehension of the problem, mainly proving lower bounds that do not suffer from a factor $K^{\gamma}$.
>
>
>
> $3^{\circ}$. Upper/lower bounds gap:
>
> Finding the minimax rates of estimation is of course a crucial objective; in that direction it would be natural to study the estimators based on the best polynomial approximation that are already known to be minimax optimal in the non-private case. Before tackling this task, we believe that it is interesting to assess the performance of simple estimators that are widely used and very easy to implement in the non private case, in order to understand the differences between LDP and non-private settings. We have thus started the study of the power sum functional with a plug-in estimator. Although such estimator  is known to be nearly minimax optimal in the non private case (up to a poly-logarithmic factor- see [14]),  we show that it performs poorly in the LDP setting (proving a tight characterization of its maximal quadratic risk, with both an upper bound and a matching lower bound). This highlights an important difference between LDP and non private setting: practical methods performing well in the non private case should not be systematically transferred to the LDP setting. As a complement, we suggest a correction of this plug-in estimator using a standard threshold method, which significantly improves the rates of convergence. We emphasize that the improvement is significant: it consists in removing the dependence of the risk in the support size $K$, when $K$ is large. This is notably different from gaining logarithmic factors, as it is often the issue in the literature on functional estimation (see the second next paragraph for sophisticated estimators gaining a logarithmic factor).
>
> We conjecture that our upper bounds are tight, up to some logarithmic factors. In future work, this logarithmic factor should be removed, using the best polynomial approximation. On the other hand, we provide universal lower bounds with respect to all privacy mechanisms and all estimators.  This gives a point of comparison for the performances of the plug-in type estimators.  Even leaving aside the logarithmic factors, we conjecture that these lower bounds are not optimal, up to a factor $K^{\gamma}$. To gain this factor $K^{\gamma}$, we may unfortunately miss a key tool in locally differential privacy, such as the two fuzzy hypothesis theorem  in the non-private case. These non-sharp lower bounds illustrate well the fact that one of the most important challenge in locally differential privacy is to provide a turnkey toolbox that help prove universal lower bounds.
>
>
> $4^{\circ}$. Comparison to the previous literature:
>
> We thank the referee for his/her suggestions. It is indeed very appropriate to cite the numerous results that have been obtained in central (or global) DP and to underline the major differences and additional difficulties related to the local DP setup. We will add such important discussion with references.\\
>
>  Updated manuscript: We have already prepared a new version of the paper that is hopefully improved by both clarifying the global presentation of our results and by slightly improving some results:
>
> (i) We have added a precise high level presentation of our results in Section 2. One theorem states the best convergence rates obtained by combining our different estimators according to the values of the problem's parameters  (support size $K$ of the underlying distribution $(p_1,\ldots,p_K)$, and power $\gamma$ of the functional $F_{\gamma}$). We discuss the phase transitions between these rates, then we outline  the rest of the paper and the relations between the sections.
>
> (ii) All the loose statements about feasible extensions and conjectures have been either deleted or clarified and re-located in the last section (discussion).
>
> (iii) We rewrote  Section 3.3 about the thresholded plug-in estimator. After having revisited the proof of Theorem 3.4, we now give a sharper error bound, which is now a minimum between two error bounds: the error bound already given (in this theorem) in the first version of the paper, and the error bound of the plug-in estimator (up to some logarithmic factor). This new bound is therefore equal to the former bound when the support size $K$ is large, and to the latter bound when $K$ is small ($K\lesssim \sqrt{\alpha^2 n}$).
> This shows that the thresholded plug-in estimator improves the performance of the plug-in estimator for large $K$, while conserving its performance for small $K$  (up to some logarithmic factor).
>
>
> (iv)  We improved the lower bound in Theorem 5.2, separating the two cases  $\gamma \in (0,1)$ and  $\gamma \in (1,2)$. For $\gamma \in (0,1)$, we kept the same lower bound as in the first version of the paper. For $\gamma \in (1,2)$, we replaced the first term  $K^{2(1-\gamma)}$ of the lower bound by $(\alpha^2 n)^{2(1-\gamma)}$. Recall that the lower bound is a minimum between two terms, where the first term corresponds to the regime $K \geq \alpha^2 n$. Since $K^{2(1-\gamma)}\leq (\alpha^2 n)^{2(1-\gamma)}$ in this regime, our replacement of the first term  is an improvement of the lower bound. This result follows from  a slight  adjustment in the proof of the theorem.

---

### Decision · Program_Chairs · 2021-09-28

**Decision:**

Accept (Poster)

**Comment:**

The paper provides LDP algorithms for estimating functionals of discrete distributions. While the paper addresses a fundamental question, reviewers raise concerns regarding the writing of the paper and the gap between upper and lower bounds. I encourage authors to incorporate reviewer concerns in subsequent versions.

**Consistency Experiment:**

NeurIPS has a long history of experimentation. In 2014, NeurIPS ran an experiment in which 10% of submissions were reviewed by two independent committees to quantify the randomness in the review process. This year, we repeated a variant of this experiment to see how the quality of the review process has changed over time.  This paper was part of the experiment and was therefore assigned to two committees (consisting of reviewers, an Area Chair, and a Senior Area Chair) that reached independent decisions.  If both committees made the same recommendation, this recommendation was followed. If a single committee recommended acceptance, the paper was accepted (with the exception of a few cases in which the other committee identified what we considered a fatal flaw, e.g., an error in a key result).

This copy’s committee reached the following decision: **Reject**

The other committee assigned to the paper recommended **Accept (Poster)**.  You can find the other set of reviews, along with any follow up discussion with the authors here:
https://openreview.net/forum?id=o1njPnYnttK